# The benefits of exploring a large scenario space for future energy systems

Ulrich Joachim Frey [1,2] ✉, Karl-Kiên Cao [1] ✉, Shima Sasanpour [1], Jan Buschmann[3] & Thomas Breuer [4]

Energy scenario analysis with optimization approaches rarely goes beyond a small number of scenarios. Disadvantages include limited coverage of uncertainties and assumptions, and a limited ability to provide robust policy advice. We present an approach that enables the multi-criterial evaluation of more than 11,000 scenarios and demonstrate it for the German power system. We vary both a wide range of input parameters and method choices. The resulting scenarios are assessed through a number of indicators on affordability, supply-security and sustainability. The most significant impacts on the results stem from considering multiple weather years. Furthermore, we estimate the number of runs required for robust energy systems analyses – well over 100 scenarios are needed. Nevertheless, fewer scenarios may be sufficient for limited scopes. Our analysis also underlines a challenge for future energy system design: cost-efficient decarbonization while conserving natural resources.

Scenario analysis is crucial for making informed decisions about future energy systems, but established modeling approaches are subject to massive uncertainties that are often not considered[1]. These uncertainties are diverse and influence various aspects of the models. First, uncertainties affect input data, relying on assumptions made by modelers. Second, each model approach only considers a few aspects of the real world. For instance, optimization models minimize costs, yet many other aspects are relevant to decision-makers. Therefore, it is important to consider a broad set of indicators for describing desired future pathways. Third, there are uncertainties related to how a certain model approach is applied, i.e., the method choices taken. One example is using different weather years or not.

Consequently, decision makers cannot consider the uncertainties caused by differences in modeling approaches, and modelers lack a systematic understanding of the effects of typical method choices. While modelers are aware that a variety of uncertainties exist, addressing every uncertainty is often impractical. Hence, it is necessary to develop a better understanding which factors have a high impact. However, uncertainty factors are often studied independently

in the literature, and comprehensive evaluations are rarely done due to the high computational effort required. Particularly in studies with only a few model runs, biases remain largely unnoticed[2]. Nevertheless, the impact of uncertainties and the influence of methodological choices on scenario results are considerable.

Energy scenario analysts typically use capacity expansion models[3], which we refer to as energy system optimization models (ESOMs). These cost-minimizing models[4] usually have normative constraints and many interdependencies. Modelers must also cope with many uncertain parameters[1]. For this reason, analyzing broad sets of scenarios is often only possible for small problems, and is established in operation planning[2] and investment planning[5] for local systems, such as residential neighborhoods[6] or industrial areas[7]. Such applications are computationally manageable since their scope is limited. However, ESOMs become disproportionately complex for research on large-scale systems, making large-scale simulations challenging. As a consequence, such ESOMs are difficult to compute[1,8–10]. High-performance computing (HPC) can address this issue and is standard in related research areas such as climate modeling. However,

[1]German Aerospace Center (DLR), Institute of Networked Energy Systems, Curiestr. 4, Stuttgart, Germany. [2]University of Graz, Department of Environmental Systems Sciences, Merangasse 18, Graz, Austria. [3]German Aerospace Center (DLR), Institute of Networked Energy Systems, Carl-von-Ossietzky-Str. 15, Oldenburg, Germany. [4]Forschungszentrum Jülich GmbH, Wilhelm-Johnen-Straße, Jülich Supercomputing Centre, Jülich, Germany. ✉e-mail: ulrich.frey@uni-graz.at; karl-kien.cao@dlr.de

although it is the best practice, it is not yet applied in many energy scenario analyses.

Presenting the results also comes with challenges. Since the purpose of scenario studies is typically to support human decision makers, the results must be summarized. Of the thousands of calculated scenarios, only a high level of aggregation or just a few representative scenarios can usually be communicated[11]. This results in a lack of incentive to calculate more than a few scenarios, given the large amount of computational resources required. In fact, even prominent large-scale energy supply studies examine only few scenarios. For example, the TYNDP 2024 Scenarios[12] consider one weather year for capacity expansion planning and three weather years for dispatch planning, as well as just two different sets of cost assumptions. Similarly, although the National Renewable Energy Laboratory (NREL) calculated 100 scenarios for their National Transmission Planning Study, only the weather years 2007–2013 were used for modeling[13].

Consequently, we must combine efficient computing of complex optimization models with systematic variations of input data assumptions to create a space of plausible scenarios, i.e., having sensible combinations of input parameters that can be communicated. In summary, we see a need for performing more robust scenario analyses that address many different system aspects. Parallel computing of many scenarios using HPC can address these needs. Accordingly, the overarching questions of this paper are: First, how can energy scenario analysis using large-scale ESOMs benefit from establishing modeling workflows in HPC environments? Second, what impact do required methodological choices have on the robustness of scenario analyses?

Our first main contribution is to inform decision-makers and non-modeling experts by raising awareness about uncertainties in the context of decision-support on future energy supply strategies. We do this by investigating the robustness of recommendations based on complex modeling approaches. To achieve this, we quantify uncertainties related to both data (parametric uncertainty) and model building (methodological uncertainty). We address these uncertainties by calculating a large number of scenarios with sampled input. In particular, this paper compares the impact of five method choices on a scenario space resulting from Monte Carlo simulations[5] for model parameters. Additionally, this paper analyses the number of runs necessary to ascertain the desired robustness of results for energy scenario analyses. Through a multi-criteria ex-post assessment, we enable different users, such as energy infrastructure planners or policymakers, to better understand if simple sensitivity analyses are sufficient or if extensive exploration of scenario spaces is necessary.

Our second main contribution is enabling other researchers to estimate the influence of five common methodological choices on results, as such systematic investigations are rare. We demonstrate this influence with more than 11,000 scenarios, thereby increasing the robustness of results and trustworthiness of such models. Finally, applying different model types can help to combine different modeling perspectives. Specifically, we couple an ESOM, REMix, with an agent-based simulation model (ABM) called AMIRIS. AMIRIS models

the main actors in the electricity market and addresses the shortcomings of ESOMs, such as the feasibility of optimal solutions in a real world with competing actors. It also allows us to analyze a broader set of indicators, avoiding blind spots, e.g., for markets, as part of a multi-criteria ex-post assessment of seven core indicators. Using this method, we obtain least-cost future power systems and test operational feasibility on the electricity market. We also assess important environmental impacts and security of supply. Although our study is based on data from the German power system, we cannot claim to provide scenario results ready for policy advice (e.g., due to the absence of up-to-date, normative policy targets). However, we provide many different methodological and parametric choices for Germany for 2030.

We present the results of a fully automated HPC workflow (see Fig. 1), which couples an ESOM with an ABM. This allows us to statistically evaluate the impact of different methodological choices on the results, the scenarios, using Python scripts. These scenarios are evaluated through a variety of indicators on affordability, supply-security and sustainability.

## Results

Energy system optimization models optimize the expansion and operation of the energy system from a central planner's perspective typically by minimizing total system costs. Optimization results heavily rely on the input data, such as weather and techno-economic data. The uncertainties within these techno-economic parameters are sampled based on a literature review, where minimum, maximum, median and mean values are identified[14]. However, optimization also depends on the methodological choices. Therefore, we evaluate the impact of five frequent methodological choices on the scenario space: (I.) Probability distribution function for input parameter sampling (truncated normal vs. uniform). (II.) Abstraction of power grids in terms of spatial resolution (small, medium, large and xl-systems). (III.) Capacity expansion approach (brownfield vs. greenfield). (IV.) Number of historical weather years considered (1 vs. 24). (V.) Network vulnerability in terms of permanent unavailability of network components (with and without).

While sometimes one option may be more realistic than its alternative – e.g., brownfield vs. greenfield – the core idea is to isolate and quantify the influence of the most important method choices in existing scenario studies. As a consequence, modelers can evaluate the influence of their method choices and, in the future, choose in a more informed way.

For each method choice, multiple sets of 100–3000 scenarios (runs) are computed (see Supplementary Table 3 for an overview). Each set represents a defined combination of the method choices (I.)-(V.). The focus is the future power supply of Germany for the target year 2030, which is modeled in two ways: as a market where interactions between decentralized actors are simulated and as an optimization of operation and investment planning from a central planner's perspective.

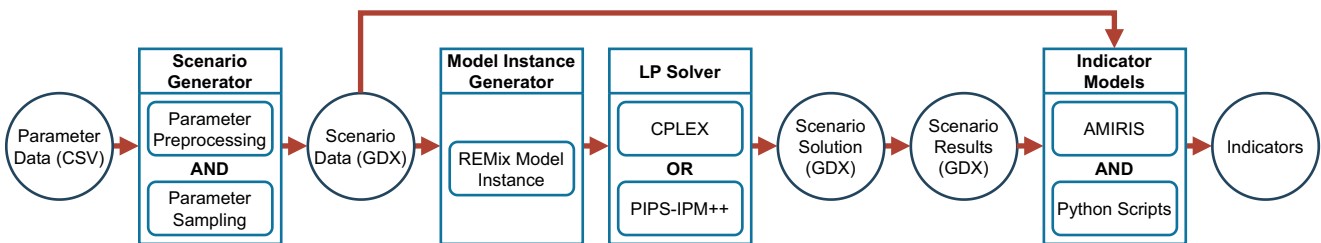

**Fig. 1 | Workflow of coupled models on high-performance computer.** The parameter sampling is done in the first step of the workflow, the scenario generator. The scenario data is fed into the model instance generator and the indicator models. The REMix results are another input to the indicator models. The indicator models calculate the seven core indicators for all considered scenarios.

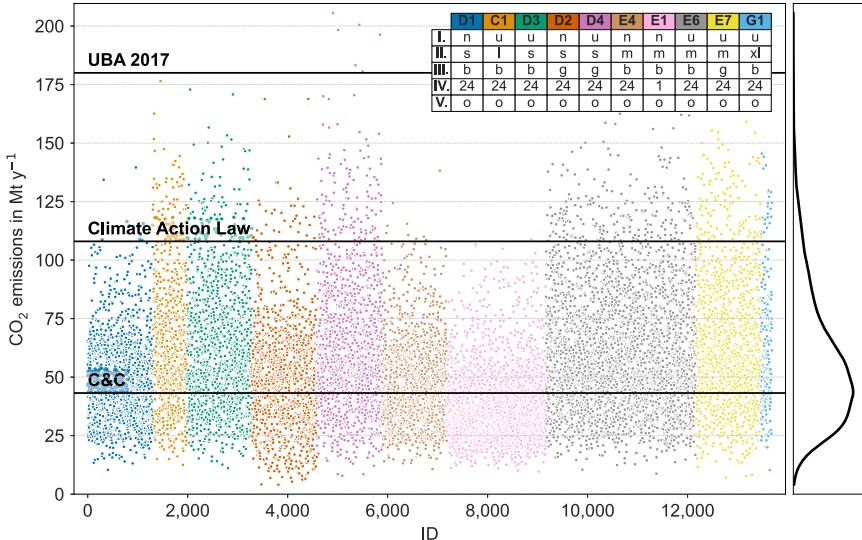

**Fig. 2 | Carbon emissions evaluated for the complete scenario space (all model instances).** Horizontal lines show different available emission budgets for reaching actual policy targets in Germany. Colors refer to scenario groups computed with a specific combination of methodological choices. On the right-hand side, the distribution of carbon emission over all scenarios is illustrated.

## Scenario space for the German power system

Figure 2 shows the resulting direct $CO_2$ emissions from our runs and the annual $CO_2$-budgets of notable scenarios for Germany as horizontal lines. Since total life-cycle GHG-emissions (provided in $CO_2$ equivalents) and direct $CO_2$ emissions are highly correlated ($p = 0.97$) across all scenarios, we treat them as interchangeable. We consider penalty costs for $CO_2$ emissions, similar to $CO_2$ emission prices used as political instrument to reduce carbon emissions[15]. Budgets for $CO_2$ emissions are often discussed in politics but, so far, they are rather used as benchmark than a strict limitation. By sampling the $CO_2$ emission price as one of many parameter uncertainties we analyze the effect of different $CO_2$ prices in combination with other parameter variations on the cost and carbon emissions for each modeling run.

Remarkably, 99.9% of scenarios stay below the UBA (German Federal Environmental Agency) 2017 budget[16], and 94% stay below the Climate Action Law's threshold[17]. These studies use a broader scope compared to our focus on the electricity sector only. According to the German Federal Environment Agency, electricity production contributed 223 Mt to the 642 Mt of emission of the energy sector in 2022[18]. Note that annual budgets in climate research decrease over time compared to the year of study, since emissions in elapsed years so far did not fall as expected. If the Contraction and Convergence (C&C) approach is applied to achieve the net-zero emission goal until 2050, the German carbon budget in the electricity sector in 2030 is limited to 43.2 Mt[19]. The C&C approach first aligns the national emission per capita globally until 2035 and then reduces the global emissions to achieve net-zero in 2050. This can only be achieved by 36% of all scenarios.

Second, for energy scenario analysis, capacities of power generating technologies are one main result. In Fig. 3, we present the differences in expanded generation capacities, if only one method choice (I.)-(V.) is switched, while all others are kept constant. Differences in method choices are calculated against the benchmark scenarios, i.e., the grey group in Fig. 2. This group is considered a benchmark because the attribute set of these scenarios − uniform distribution of inputs, brownfield approach, medium sized grid resolution, 24 weather years, no grid outage − resembles a robust combination of modeling choices the most.

As can be seen, the probability distribution (I. = normal) has the biggest impact on the structure of the energy system. With a normal distribution, significantly less capacities are expanded. A greenfield approach (III. = green) leads to a shift from wind to photovoltaics power plants. In contrast, spatial resolution (II. = large) has a negligible impact on infrastructure requirements. Just considering one weather year (IV. = weather) results in an energy system that might be insufficient for years with less hours of sunshine and wind. If particular network nodes in the transmission grid are considered to be unavailable (V. = outage), more power generation technologies (especially photovoltaics) are installed.

## Methodological choices

To get a deeper understanding of the differences resulting from methodological choices, we use seven core indicators considered as most relevant for future energy systems: System costs, (life-cycle) greenhouse gas emissions (in $CO_2$ equivalents), maximum energy not served, average electricity price, land use, dissipated water, and minerals and metals required. They cover the essentials of the energy supply–sustainability, affordability and system security. More details can be found in ref. 20, as well as Table 1 and Supplementary Table 4.

Figure 4 shows the normalized distributions of the indicators resulting from our scenario generation approach. Within Fig. 4, each method choice is represented by one subplot comparing two scenario ensembles which are characterized as shown in the legends. The attributes (in the legend) stand for a different method choice, respectively.

## Parametric uncertainty of core indicators

In particular, two indicators – land use and maximum energy not served – already show a large range of values independent of method choice. For the latter, this is due to the selected modeling approach, which assigns high penalty costs to supply interruptions and thus, ensures a high level of security of supply for most scenarios. However, a few scenarios show very significant supply interruptions, resulting in a long tail of values (see Fig. 4f). The system cost indicator, which is minimized, shows the smallest range of values. Uncertainties regarding environmental impacts and electricity prices vary more widely, as does the indicator for greenhouse gas emissions (GHG).

## Changing the probability distribution function (I.)

Previous scenario research has shown that results heavily depend on the assumptions of input variables[21,22]. Instead of trying to get to more

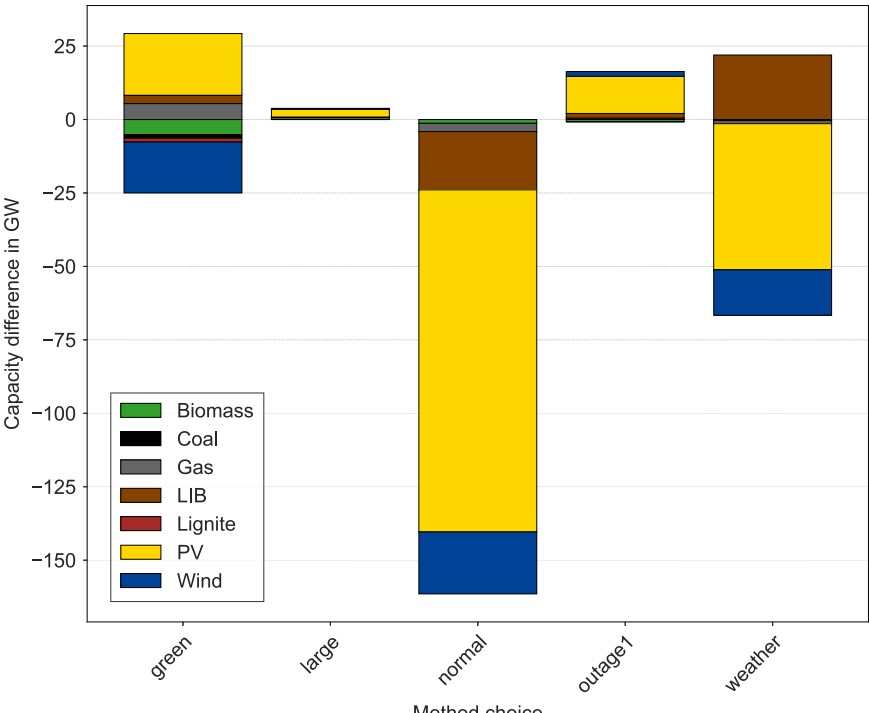

**Fig. 3 | Mean capacity differences for different methodological choices compared to reference case.** The reference case (E6) consists of a uniform distribution, medium-size instance, brownfield optimization, 24 weather years, and no network unavailability. The capacity differences are shown for model instances where only the mentioned methodological choice differs from the reference case. A normal instead of a uniform distribution for the parameter sampling results in significantly lower total capacities. The impact of the spatial resolution is comparatively small. A greenfield optimization results in more expansion of closed-cycle gas turbines, photovoltaics and lithium-ion batteries. Wind power plants are expanded to a lower extent. Considering only one weather year might result in an undersized energy system. Network unavailability results in more decentral power generation.

specific values, we draw from distributions of 80 scalar input parameters based on a literature review. This review contains real scenario studies that do not focus on generating extreme values. Thus, the given maximum/minimum values are actual extremes within the screened literature, suggesting that still more extreme scenarios are conceivable but not very plausible. Two distributions, truncated normal and uniform, are investigated.

If a normal distribution instead of a uniform distribution is chosen, keeping all other methodological choices the same, significantly less power generation capacities are expanded. Power generation decreases with few exceptions (closed-cycle gas turbines +31.6%). As a result, less $CO_2$ is emitted (Fig. 4a). Due to the smaller variance of the truncated normal distribution, the spread of the system costs is smaller compared to the uniform distribution. Overall system costs are approximately 25% lower with a truncated normal distribution while other core indicators also exhibit smaller values (see Supplementary Table 5).

The reason for this is a larger total electricity demand as an input parameter: The value for Germany's total demand results from the summation of the demands across all network nodes. The nodal demands are subject to parameter sampling and thus, are directly influenced by the selected probability distribution. The truncated normal distribution of the annual demand in Germany is skewed to the right, i.e., higher demand values have a smaller probability (see Supplementary Table 1). As a result, with a mean of 715.8 TWh, we observe much greater total electricity demands when using uniform distributions compared to 636 TWh in case of sampling with truncated normal distributions.

In summary, the use of truncated normal distributions leads to a less extreme scenario sampling. Hence, more scenarios meet decarbonized future energy pathways in a cost-effective manner.

## Impact of abstracting the power grid (II.)

To keep computational times at a manageable level, model size is often adapted by varying the temporal or spatial resolution. Hence, we vary this key factor for German scenarios as well, between a small (fully resolved only for one part of the high-voltage grid in North Rhine-Westphalia and aggregated for the rest of Germany), medium (fully resolved only for the high-voltage grid in North Rhine-Westphalia), large (high-voltage grid highly resolved in half of Germany) and a very large model (high-voltage grid of Germany fully resolved). Resulting computational times vary dramatically. While calculating 1300 small models took just 92,000 core hours on average (the 5400 models actually calculated are at 380,000 core hours), the 1300 medium ones are already at 280,000 core hours on average (11,700 models actually calculated are at 2.3 million core hours), the large ones would be at 12.8 million core hours (820 models actually calculated are at 2.9 million core hours) and the very large models would use up 13.7 million core hours for 1300 runs (300 models actually calculated are at 3.2 million core hours). An overview of the computational times and data usage is shown in the Supplementary Table 3.

The spatial resolution of model instances has a rather small impact on the optimization results. If the spatial resolution increases, some wind offshore power plants are replaced by more decentral wind onshore power plants. The grid-related curtailment increases and therefore, gas-fueled power plants are dispatched to a higher extent. At the same time, $CO_2$ emissions and the total system cost increase. Furthermore, the spatial resolution impacts the maximum energy not served and water use (see Fig. 4b).

Of these indicators, only system cost is highly significantly different, with high spatial resolution scenarios having a much higher overall cost for an optimal system (see Supplementary Table 5). The reason behind that is simple: The coarser the spatial resolution, the

**Table 1 | Selection of most relevant indicators for assessing affordability, security and sustainability**

| Indicator | Reported unit | Description |
|---|---|---|
| System costs | 1000 € | The optimization is performed by minimizing the overall economic expenses associated with the operation, maintenance, and infrastructure development of the energy system. These expenses include fuel procurement, power generation, transmission, and energy storage costs. |
| (Life-cycle) GHG emissions | 1000 t | This indicator refers to the total amount of emissions of $CO_2$ equivalents generated by the power system. If not indicated otherwise, we report the life-cycle greenhouse gas emissions that include all stages of the life cycle, from raw material extraction and processing, manufacturing, transportation, and use, to disposal or recycling. |
| Maximum energy not served | GW | The balance between energy production and demand in the power system is crucial for ensuring energy security and reliability. In the optimization, a slack variable is employed to supply energy at a very high cost when there are no other economically feasible options for delivering electricity. The maximum value of this indicator signifies the extent and timing of potential threats to the energy system. |
| Average electricity price | € MWh$^{-1}$ | This indicator represents the electricity price at the energy exchange, weighted by the amount of energy, and averaged over one simulation year. It is directly related to household and industry electricity prices and serves as an indicator of consumer costs for electricity. |
| Land use | Dimensionless, aggregated index[45] | Land use denotes the extent to which land is utilized by energy generation plants, taking into consideration potential conflicts with other land uses, such as agriculture. Accordingly, low land use is desirable for a sustainable energy system. |
| Dissipated water | m$^3$ water$_{eq}$ | Dissipated water refers to the water requirements during the production and operation of the energy system. With increasing concerns related to climate change and declining groundwater levels due to excessive use, water resources have become increasingly important. |
| Minerals and metals | kg Sb$_{eq}$ | The depletion of minerals and metals reflects the total resource consumption associated with the construction and operation of power plants across all types of minerals and metals (in kg antimony-equivalents). |

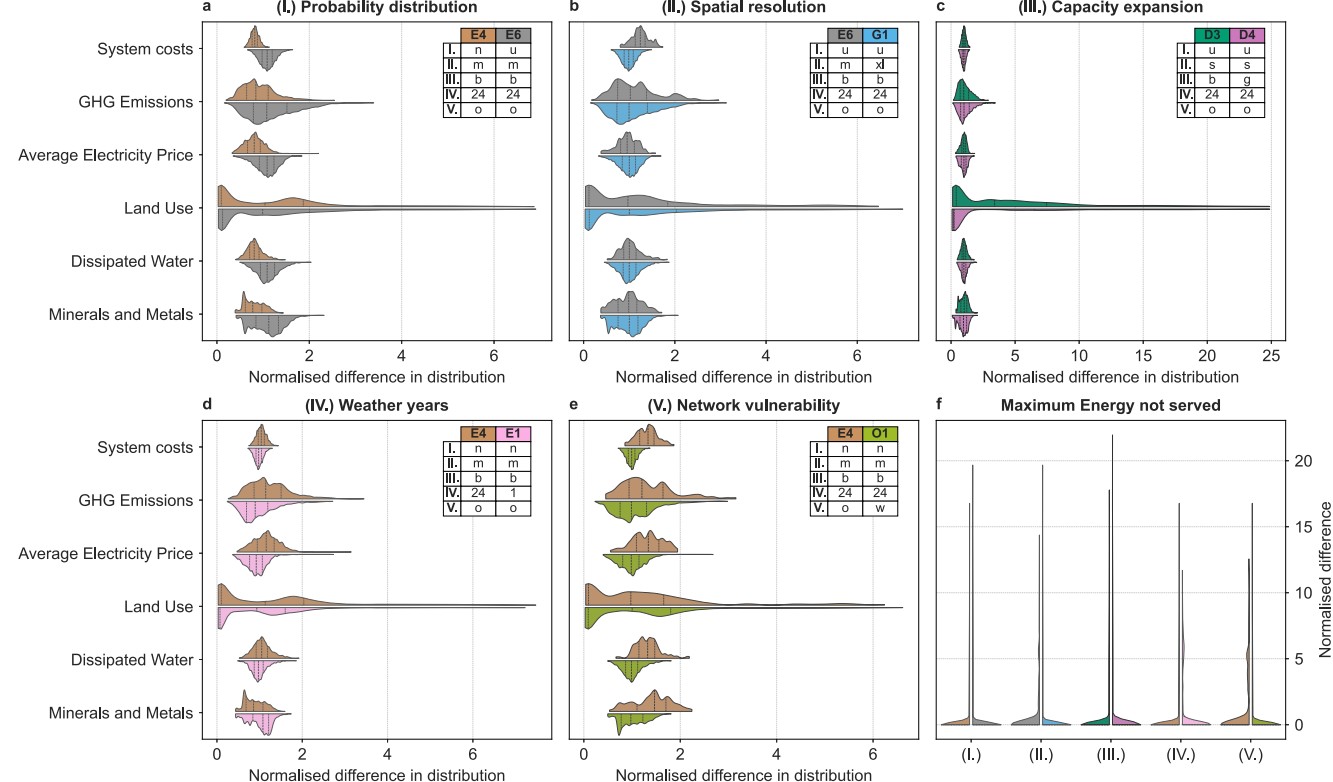

**Fig. 4 | Comparison of key indicators for different method choices.** For each subfigure and indicator, the median value over both scenario groups is calculated and the plot is normalized accordingly. The dashed lines represent the median value of each scenario group and indicator. **a** Probability distributions. A uniform distribution results in higher system costs and higher greenhouse gas emissions. **b** Spatial resolution. The impact on most indicators is rather limited in comparison to other methodological choices. **c** Capacity expansion approach. A greenfield approach results in lower system costs compared to a brownfield approach.

However, the carbon emissions increase in average. **d** Number of weather years. Considering 24 instead of only one weather years has a significant impact on all considered indicators. **e** Unavailability of network components. Considering network unavailability leads to reduced carbon emissions since decentral renewable energy technologies are utilized to a higher extent. **f** Impact of the five method choices on the maximal energy not served. Few scenarios show very significant supply interruptions. See Supplementary Table 6–10 for the respective t-tests.

smaller the model, hence less transmission lines are considered. They can be subject to congestion and thus either make investments into grid infrastructure or power generation units at sub-optimal locations necessary. Similar observations have already been made in Refs. [23] and [24].

However, a remarkable finding in this study are the insignificant differences of the remaining indicators. Accordingly, simple models are already sufficient to develop useful estimates of the performance of energy scenarios, e.g., with regard to their sustainability.

### Changing the capacity expansion approach (III.)

Although we are looking at scenarios for 2030, which speaks in favor of a brownfield approach for existing power plant capacities, a greenfield approach enables the modeler to analyze more extreme scenarios. While a brownfield approach presents a more realistic consideration of preexisting capacities, a greenfield approach can show optima that are hidden behind today's system structures[25].

With a greenfield instead of a brownfield approach, the number of conventional coal, lignite and gas power plants is reduced below the capacities that are available today. These missing power plants are compensated by additional closed-cycle gas turbines (CCGT), photovoltaics power plants, and further lithium-ion batteries to balance the generation peaks of photovoltaics. However, on average, about 25% less wind onshore capacities are available, since photovoltaics power plants are a cheaper zero-emission alternative for Germany. This also means that in around half of the greenfield scenarios less wind onshore capacities are available than today. Since photovoltaics yield is highly time-dependent compared to onshore wind, an alternative in the form of CCGT and battery storage is required for sunless hours of the day. Therefore, while the total costs decrease since more degrees of freedom are available, compared to the brownfield approach, $CO_2$ emissions increase.

The results for the other indicators are as expected: Since the most cost-efficient technologies can be built, greenfield systems tend to be less costly and are more desirable in terms of sustainability indicators. For example, open-cycle gas turbines are replaced by closed-cycle gas turbines. Figure 4c shows the significant difference in land use for small-sized model instances. The same holds true for medium-sized model instances, making the results robust, since both directions and significances are almost identical for small and medium instances for all variables that are highly significant (see Supplementary Table 8). However, security of supply is worse in greenfield systems, since there is less need for thermal power plants providing firm capacity. Accordingly, costly uncovered demand, as expressed by the indicator maximum energy not served, is accepted by the optimizer, if particularly high specific investment expenditures result from the parameter sampling.

### Weather years (IV.)

Despite the common knowledge that using a large set of historical weather years is necessary, this is still not common practice. Since weather-dependent renewable energies will increase and finally dominate the energy system, this is a severe limitation, limiting the robustness of results. The only exceptions are studies in which the effects of changing weather time series are specifically investigated by means of sensitivity analysis[26]. The distinctive aspect of our analysis is the use of a large set of weather years combined with additional parameters. Therefore, our fourth method choice compares scenario ensembles for a single weather time series with an ensemble that samples from 24 different weather years (1995–2018).

If only a single weather year is used for the optimization, the resulting energy system might be undersized. Since the historical weather year 2018, which we use for our analysis, was a particularly good one in terms of solar power, significantly less photovoltaics and wind capacities are necessary. For balancing purposes, more lithium-

ion batteries instead of biomass and CCGT power plants are expanded. However, if several weather years are considered, the size and structure of the energy system differs significantly. Lombardi et al. also identify weather variability as having the highest sensitivity for the results when comparing different uncertainties[27]. Schlachtberger et al. analyzed the impact of weather year uncertainties as well[28]. While this study also observes differences in the size and structure of the energy system when comparing four different weather years, the effect is not as prominent in comparison to our analysis of 24 weather years. This emphasizes the importance of a broad consideration of weather years.

Remarkably, there is also a highly significant difference for all indicators, e.g., the mean electricity price (Fig. 4d). As can be seen in the Supplementary Table 9, all differences are significant at the $p < 0.001$ level, with more weather years associated with more $CO_2$-emissions and higher system costs. These results gain in importance when considering that the standard is just one weather year.

### Network vulnerability (V.)

The capacities resulting from least-cost optimization approaches represent a lower bound in terms of system adequacy[29]. Consequently, energy scenarios should account for cases when certain components of the energy system are not available due to planned (e.g., power plant revisions) or unplanned outages (e.g., due to an attack). In this section, we test how the common approach of simply subtracting one element from the system ($n$-1) influences scenario results (Fig. 4e). Therefore, we define five network nodes in the transmission grid that could be affected by outages. As a result, transmission capabilities for electricity are more limited. This is why decentralized technologies such as photovoltaics and onshore wind are being expanded and utilized more intensively. As a consequence, conventional power plants are dispatched less and $CO_2$ emissions decrease on average. We observe the same trends for each of our key indicators. However, these effects are not significant (see $p$-values in the Supplementary Table 10).

Note that we only consider permanent outages of network nodes without generation units. Extending outages to generation units would result in the trivial result that these missing units would just be replaced by other, additional units. Hence, temporal outages and testing for the unavailability for each of the network nodes may lead to more significant observations.

### Minimum number of runs needed for unbiased results

Apart from methodological choices as shown above, the number of runs needed for robust, trustworthy results is highly relevant for modelers. This number has a direct impact on the computational resources required which, in most cases, are the limiting factor. Estimates range from one hundred to several hundred runs, but the number of scenarios is often said to be mainly dependent on the desired confidence level[1].

To assess the impact of sample size variations, the largest scenario set with 3000 scenarios is set as benchmark by calculating the means of our seven key indicators. A random sample of 100 scenarios is drawn from the pool of 3000 scenarios, where the sample size increases from 100 to 2900 for each draw. Each random sample (for each size of 100–2900) is drawn 100 times to ensure robust draws and eliminate bias.

In Fig. 5, we show how the indicator values of each sample differ compared to the mean value of all 3000 scenarios. Each colored line represents the difference of a sample mean to the overall mean of 3000 runs. The shadowed area shows the differences for the upper and lower quartile. This gives us an accurate picture of the error made by evaluating only small scenario sets:

As can be seen, for most indicator deviations lie between 0 and 5%. This is different for the land use indicator (beige) and the maximal energy not served (blue). For these indicators, a difference of less than 5% is reached with a scenario set size of about 400 and 1000 scenarios,

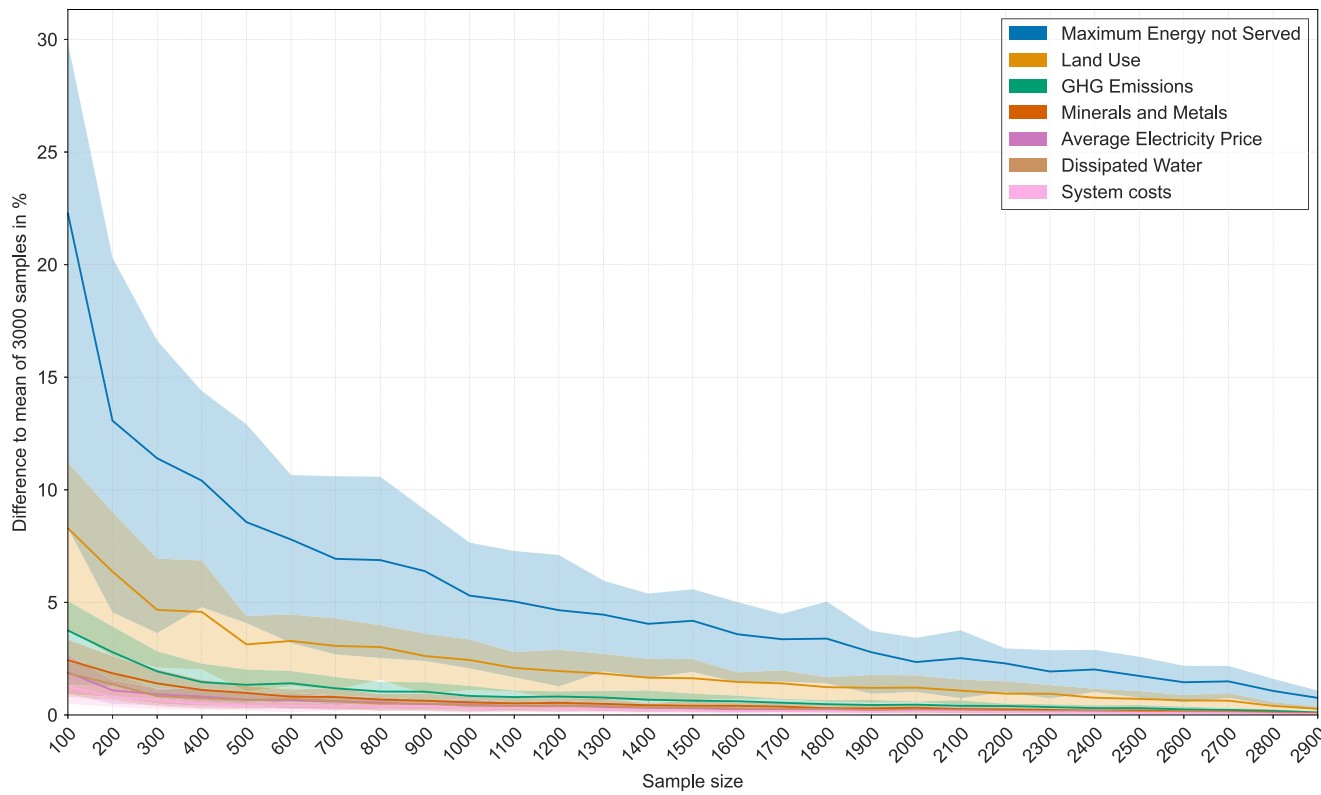

**Fig. 5 | Relative deviation of key indicator values compared to the mean value for 3000 scenarios.** Deviation of the seven key indicators for differently sized scenario samples where each random sample is drawn 100 times to ensure robust draws and to eliminate bias. For most indicators a sample size of 100 results in difference of less than 5%. However, for the indicators "land use" and "maximum energy not served" at least 400 and 1000 scenarios need to be calculated to achieve a difference of less than 5%.

respectively. However, the largest error decrease is up to a set size of 400 with an underestimation of the energy not served of about 9%.

Hence, as long as up to 5% deviation for a majority of key indicators and no more than 10% deviation for the maximal energy not served is acceptable, a lower boundary of just 400 scenarios might just be enough, if computational resources are scarce. However, this precision may not be enough for some applications. Since typically a few hundred of scenario runs are deemed to be enough, this supports the existing literature[1]. Note that these approaches determine the number of runs statistically, calculating the 95% confidence interval but do not explicitly present distributions of indicators like in Fig. 4.

## Discussion

In this paper, we emphasize the importance of analyzing the impact of different method choices on model results, as they are a source of uncertainty. We assess these results using multiple indicators beyond costs in energy systems analysis. A truly optimal system performs well in at least a few indicators. However, one method choice we keep constant is the system cost minimization, which affects the observed variation of this core indicator. Hence, for each scenario we obtain the best possible performance for this indicator. However, we find some systems that perform well in 4 out of 7 indicators.

It is also possible to calculate Pareto fronts, as Fig. 6 shows, where we depict the popular trade-off between system costs and greenhouse gas emissions. It confirms previous findings in the literature: Close to the cost-minimum exists a large spectrum of scenarios that perform very heterogeneously in terms of other indicators, such as GHG emissions. Remarkably, systems that perform well in terms of minerals and metals use are neither close to the system cost minimum nor the GHG minimum. Scenarios that serve desirable trade-offs between

these divergent indicators can be determined using multi-criteria optimization. However, this method is computationally costly, which underlines the need for further analyses such as this.

There are some limitations of our study. First, the analyzed method choices represent only some of the decisions made by modelers. Further candidates are the degree of abstraction for generators and storage units, or the impact of climate change (e.g., on demand patterns). In addition, the analysis of parameter variations can never be complete.

Second, since the consideration of sector integration was out of scope, we can only hint at their impacts. For example, relying more on variable renewable power generation, considering multiple weather years becomes even more crucial. In contrast, stronger interconnectedness of energy sectors may strengthen a system's resilience against individual outages. More detailed research in this direction becomes especially relevant in highly decarbonized future energy systems, for which sector integration is even more important. Given the high adaptability and modularity of the models, the workflow could be adapted for scenarios with sector integration or other specifications.

A third limitation of the modeling setup is that most scenarios have a high resolution for one German federal state only (North Rhine-Westphalia) and a lower resolution for the rest of Germany and neighboring countries. A fourth limitation, as shown in the results overview, the scenario space does not include scenarios with a share of 100% of renewables. However, we are well aware that there are more ambitious scenarios[19]. Given abrupt changes in policies, market-related indicators have been calculated without policy variations. Evaluating different policy regimes is an obvious next step, alongside the consideration of sector integration in a fully-decarbonized energy system.

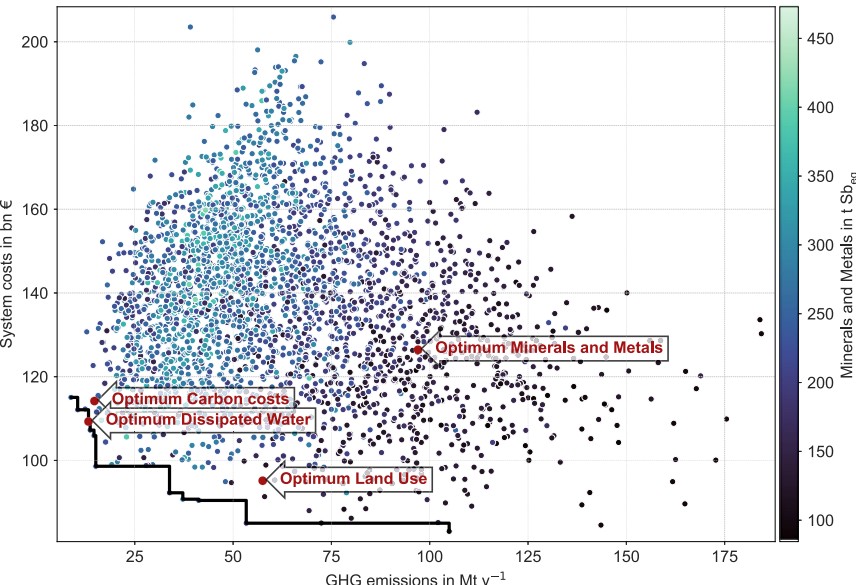

**Fig. 6 | Pareto front of greenhouse gas emissions and system costs for scenario ensemble E6 (uniform distribution, medium-size instance, brownfield optimization, 24 weather years, and no network unavailability).** Every dot is one scenario and the colors show the indicator for utilization of minerals and metals from dark blue (less consumption) to light blue (more consumption). The red dots show the best performing scenario for each indicator named.

Finally, we did not consider any spatial interrelationships of sampled parameter values. This mainly affected independently sampled annual demands for every network node and can be seen as implicit assumption: Annual demand of different regions develops into opposite directions, which is only partially plausible (e.g., due to urbanization). Endogenously modeling energy demands may resolve this limitation. Besides such improvements, the robustness of energy scenarios may be increased by integrating the target triangle of energy supply into the ESOMs, i.e., via multi-criteria optimization of indicators. As a use-case, we study the future German power system for which we use greenhouse gas emissions, system cost, the maximum of energy not served, land use, the use of minerals and metals, dissipated water and the resulting average electricity price as key indicators for scenario assessment. This also enables us to statistically quantify the benefits of numerous scenarios by comparing ensembles of different sizes. Our findings highlight the importance of using a large variety of empirical time-series data that reflects the uncertainty of renewable power feed-in. We find that all key indicators are significantly impacted by this parameter.

System costs are significantly affected by four of the five exemplary methodological choices ($p$-values < 0.001 for each, parameter sampling approach, abstraction of grids, capacity expansion approach, number of weather years), while the observed coefficient of variation for greenhouse gas emissions is 48.3% for the full set of more than 11,000 scenarios. This variation is even greater for the land use sustainability indicator (124.2%), demonstrating the need for more deliberate communication of the uncertainties involved.

The results demonstrate the importance of analyzing large ensembles of energy scenarios to answer questions on $CO_2$ abatement costs and security of supply. These questions are relevant, for example, for generation capacity adequacy assessments. In numerical terms: We observe large deviations in the security of supply indicator of up to 30% for small scenario ensembles compared to large ones. Therefore, studies emphasizing the determination of methodological choices require well-founded assumptions.

For energy system modeling experts, we demonstrate how to systematically evaluate large ensembles of energy scenarios using workflows on a high-performance computer. The workflow we have implemented represents an important step forward in overcoming the current practical limitations that prevent energy scenario analyses today from providing the same high-quality insights as climate research. In this context, harnessing HPC becomes even more important because for real-world policy support, our workflow would need to incorporate even more data, such as normative policy targets or coupling of all energy sectors.

For non-modeling experts, we demonstrate that analyzing numerous scenarios is crucial for any study on future systems where uncertainties might significantly impact the optimization results. Additionally, we highlight key issues to consider when making decisions based on typical modeling frameworks in the energy sector. Using indicators that represent the target triangle of energy supply – affordability, security, and sustainability – the impact of the uncertainties considered is measured.

## Methods

This section describes the complex setup in detail: a parallel solver used within an automatized workflow coupling different energy system models using High-Performance Computing.

The focus of our investigation is the future power supply in Germany. In order to achieve a broad coverage of all essential aspects, we coupled two complementary models. The first, REMix, is an ESOM, the second, AMIRIS, is an agent-based simulation of electricity markets. Accordingly, we model the German power system as an optimization of operation and investment planning from a central planner's perspective and as a market where interactions between decentralized actors are simulated. The next two sub-sections shortly introduce both models.

### Energy system optimization model

REMix (Renewable Energy Mix) is a framework for energy system optimization models (ESOMs). Different sectors, like power, heat and transport as well as several technology groups (power plants, storage and transport technologies) can be modeled within the optimization. For this study we use it for linear optimization of one target year with hourly resolution and perfect foresight. However, REMix provides further features like path optimization, mixed-integer programming and multi-objective optimization[30]. Besides, the possibility to automatically perform a spatial aggregation of input data, simple temporal

model reduction (down-sampling of times series) is possible, too. The input data includes weather and demand profiles as well as techno-economic parameters like technology-specific capital and operational expenditures, energy conversion efficiencies and life times.

Within this study, we focus on the power sector and therefore our network model of Germany has a high spatial resolution, where each node represents a transformer substation, taking imports and exports to neighboring countries into account[31]. The total number of 488 nodes can be spatially aggregated in order to reduce the model size and computing time of the model. For the reference case, only the federal state of North Rhine-Westphalia is highly resolved and the rest of Germany is aggregated. This results in our medium-sized model instances consisting of 109 network nodes. The technological focus is on the power sector with renewable and conventional power plants, the electricity grid, battery and pumped-hydro storages considered. Carbon emissions are penalized with additional $CO_2$ costs, which are subject to the scenario sampling (see Section Scenario generation and parameter sampling). For the power demand we use historical load profiles for Germany from ENTSO-E. The future annual demand is based on a literature review and disaggregated to the model nodes according to population density. It considers conventional power demand as well as power demand from other sectors (e-mobility, heat pumps etc.).

AMIRIS (Agent-based Market model for the Investigation of Renewable and Integrated energy Systems) is an electricity market simulation model for Germany that incorporates many different agents with complex decision strategies[32]. Its focus is on assessing the impact of energy policy instruments on the economic performance of power plant operators and marketers[33]. It has an hourly resolution and computes electricity prices based on the bidding behavior of proto-typed market actors based on the FAME framework[34]. It is calibrated on historical data. Different groups of actors are represented as individual agents with varying degrees of uncertainty and limited rationality, reflecting their heterogeneity. For example, renewable energy mar-keters are defined in detail by their portfolios of contractually linked renewable energy plants, cost structures, price and performance forecasts, and capital stocks. Support instruments like market premia are also considered, influencing the decisions made by actors involved in renewable energy sources, such as plant operators and their mar-keters. Regulatory frameworks governing these market interactions, e.g., on the spot market and control energy market, for instance, are calculated as well. One example are costs of technologies like feed-in tariffs.

AMIRIS as used here is parametrized using the calibrated data for Germany (https://gitlab.com/dlr-ve/esy/amiris/examples), with no explicit policy modeling. Its main output, the mean electricity price as well as all other market indicators, are calculated via a post-processing script of REMix results. AMIRIS serves two main purposes. First, it acts as a plausibility check for optimized energy systems as calculated by REMix, which may be optimal in a macro-economic, but infeasible from a micro-economic perspective. Actors will act for their own economic benefit, which can be represented by AMIRIS. This means that in comparison to REMix, the actors in AMIRIS optimize their own profits without taking the optimal operation of the energy system into account. Therefore, AMIRIS can be applied to better represent the electricity market[35]. Second, it adds the market driven indicators to the indicator set, e.g., electricity prices.

## High-performance computing workflow

In order to be able to run thousands of scenarios while systematically and automatically varying input parameters and methodological choices, a fully automated workflow had to be developed on a high-performance computing (HPC) facility, JUWELS[36] and JURECA-DC.

For this, a tool chain of coupled models and software packages had to be linked allowing us to conduct numerous large-scale scenario analyses. The model chain (Fig. 1) includes scenario generation, energy system optimization using ESOM instances created with REMix[30] and mainly solved with PIPS-IPM++[37], agent-based market simulation through AMIRIS[32,38], which is based on the FAME framework[34] and FAME-IO[39], and further post-processing for evaluation of results in terms of a multi-dimensional indicator set. For an automatized and systematic execution of this tool chain we used JUBE[40], a script-based framework for running and benchmarking complex workflows. Finally, the statistical evaluation of the resulting scenario space has been done in R[41] using a desktop PC.

Within our interdisciplinary collaboration, the software packages employed were developed by distinct teams and had not previously been integrated to yield scientific results on an HPC system. As each component is a sophisticated tool in its own right, achieving a seamless workflow required meticulous planning and precise interface design. Moreover, effective communication among team members was indispensable to maintain overall system coherence. Technical implementation details are provided in ref. 42.

## Scenario generation and parameter sampling

The parameters, which are the basis for the workflow, are subject to uncertainties. In other words, for robust scenario results, the inputs of the model instances are crucial. Hence, we conducted an extensive literature research considering approximately 50 literature sources and derived statistical descriptors (min, median, max) for a selection of important parameter values to be varied. This resulted in an extensive parameter space[14].

The considered parameter variations include fuel cost, invest-ment cost, fixed and variable operation and maintenance (O&M) cost, life- and amortization time, efficiency, annual biomass potential and annual demand uncertainties (see Supplementary Table 1). The annual demand considers the power sector demand and the electricity demand from road transport and the heating sector. The annual bio-mass potential and the annual demand are disaggregated to the con-sidered model nodes according to today's biomass and population distribution, respectively. Additionally, the historic weather years from 1995–2018 are considered. The normalized historic load profile from 2006–2015 can be sampled independent from the annual demand.

Furthermore, we conducted interviews with experts in the field of energy system analysis on the interrelations of all parameters within the parameter space in order to derive a quantitative pseudo-correlation matrix consisting of integer values ranging from -3 for strongly anti-correlated to +3 for strongly correlated. For this, first an empty matrix had to be filled by experts to identity the non-zero interrelations between the parameters to be varied. Subsequently, in a moderated survey (questionnaire), both online and in presence, the assessment of the non-zero interrelations has been revised pairwise. The dominating answer finally defined the interrelation value that was selected. The matrix was required for plausible parameter sampling results, e.g., to avoid the minimum cost for oil occurring simulta-neously with the maximum cost for natural gas, considering that these fuel prices are highly correlated today. Therefore, the pseudo-correlation matrix also ensures that independently random sampling of certain parameters (interrelation factor 0) is an explicit decision rather than implicitly decided by ignorance. Both the parameter space and the pseudo-correlation matrix have been fed to a tailor-made scenario generator that automatically sampled parameter sets for the subsequent workflow steps. The parameter sampling approach has been applied to all scalar values of the parameter space and could vary between uniform and truncated normal probability distributions. For example, Kang et al. assume a normal distribution for cooling demand and a uniform distribution for the gas price[43]. In our case, a truncated normal distribution is created to assign a lower probability to extreme values. For the expectation of the truncated normal distribution the median value of the researched parameters is used. We use the range

rule of thumb which approximates the standard deviation $s = (\max - \min)/4$. Then, the normal distribution is cut off at the minimum and maximum value to receive a truncated normal distribution. However, since the development and therefore the actual probability distribution of future parameters is unknown, the truncated normal distribution is compared to a uniform distribution, where all possible values have the same probability.

There were two exceptions to the parameter sampling described above: First, weather data which has been provided as spatially resolved time series of potential power generation from renewable energies derived from the COSMO-REA6 data[44]. Here, we always used a uniform probability distribution to randomly pick one out of the 24 weather years available. Second, the unavailability of network nodes has been conducted by constructing five violated network topologies, where passive network nodes (without power generators or demands assigned) have been manually removed together with their links. Each of these outage topologies has been tested for the first 100 similar parameter sets created with the scenario generator.

We deliberately decided not to use scenario reduction techniques because scenario reduction leads to undesirable results. Ideally, reducing the number of scenarios by 80% would retain 80% of the important information. However, in our case, which should be rather representative of scenario runs in general, the reduced scenarios contain only 10% of the information. This is because our sampling approach treats different parameter types differently: Time series are drawn as complete data set and contain several thousands of numerical values. Opposed to the comparably small number of individually sampled scalar values (i.e., techno-economic data) their impact on reduced scenarios would dominate. In other words, scenario reduction would simply combine scenarios with the same selected time series and marginalize variations of techno-economic data.

Furthermore, in case of very large model instances about 20% of the started runs crashed. We do not fully understand which circumstances cause this failure and thus cannot exclude the possibility that a specific group of parameter combinations is the reason or a numerical instability in the solver at a very large scale. Therefore, we may not be aware that a certain part of the parameter space is not evaluated for large model instances.

### Spatial aggregation

The spatial aggregation is based on clustering according to electrical distance, to keep the most likely grid bottlenecks in the model. Parts of the network are aggregated in clusters while the rest remains in the original spatial resolution. We chose to model a technology-rich and highly interconnected region within Germany with high spatial resolution. This allows us to analyze different types of conventional power plants and their phase-out more precisely. Furthermore, the highly resolved part enables us to consider a set of different security of supply indicators and the unavailability of transformer substations. Aggregation directly sums up power generation and demand within a cluster, ignoring possible transmission constraints. For method choice II., spatial abstraction of the German extra-high voltage grid, four different model sizes are defined: small (grid in a small part of Amprion's region fully resolved ($n = 44$; the rest of Germany aggregated to 10 model nodes), medium (grid in North Rhine-Westphalia fully resolved ($n = 99$; rest of Germany aggregated to 10 model nodes), large (grid in half of Germany fully resolved ($n = 264$; the other half of Germany aggregated to 6 model nodes), very large (grid of Germany fully resolved ($n = 488$)).

Resulting computing times vary dramatically. While calculating 1300 small models took just 92,000 core hours on average (5400 actually calculated are at 380,000 core hours), the 1300 medium ones are already at 280,000 core hours on average (11,700 actually calculated are at 2.3 million core hours), the large ones 12.8 million core hours (820 models actually calculated are at 2.9 million core hours)

and the very large models would use using up 13.7 million core hours for 1300 runs (300 actually calculated are at 3.2 million core hours). In total, this workflow produced millions of files (40 TB) in more than 366,000 directories.

### Capacity expansion approach

In our case, the existing power plants are based on the installed capacities in Germany in 2018. Additionally, we take the lifetime of these power plants and political decision like the phase-out of nuclear or coal-fired power plants into account. In the brownfield approach, capacities of remaining power plants are from 2018. Further capacity expansion is only possible for renewable and natural gas-fueled plants. In contrast, the greenfield approach does not consider any preexisting power plant capacities. All power plants can be expanded; however, the maximum capacity of coal-fired plants is restricted to the existing capacity in 2018, similar to the brownfield approach.

### Indicators

Table 1 describes the selected core indicators. In order to broaden our analysis of the scenarios we selected seven indicators that assess different aspects of affordability, security and sustainability. Further details are provided in ref. 42 along with a much more comprehensive list of indicators tested for our experiments.

## Data availability

The scenario data generated in this study have been deposited in b2share under accession code 7dfe93339c3e4e34bf4c47f880186466. The model instances generated in this study have been deposited in b2share under accession code 3717dab82cbb4de0a02726ab3ff7702e. The techno-economic data used in this study are available in b2share under accession code (4e5e2d11b8224fb8809cdc2d07eeff04). The used modeling frameworks REMix and AMIRIS are published in JOSS at (https://doi.org/10.21105/joss.06330) and (https://doi.org/10.21105/joss.05041), respectively. The codes to obtain the REMix basic model are published in Gitlab (https://gitlab.com/dlr-ve/esy/remix/projects/powger).The codes for determining the indicators, the scenario generator tool and the configuration for controlling the HPC workflow with jube are published in Gitlab (https://gitlab.com/dlr-ve/esy/remix/projects/unseen). Data that supports the figures and other findings of the study are provided in the Supplementary information (SI). Further data, such as sampled model inputs and intermediate results are maintained at the Jülich Supercomputing Centre and can be made accessible after registration in the JuDoor portal given the size (40 TB) of the data sets generated.

## Code availability

All individual components of the HPC workflow are published open source[30,32,34]. PIPS-IPM++ is available on (https://github.com/PIPS-IPMpp/). The JUBE-software is openly available[40]. The work flow manager ioproc is available at (https://gitlab.com/dlr-ve/esy/ioproc). The code for running the HPC workflow, and parts of the indicator processing is available on (https://gitlab.com/dlr-ve/esy/remix/projects/unseen). The code for the statistical evaluation will be made available upon request.

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

## Acknowledgements

The research was funded by the German Federal Ministry for Economic Affairs and Energy under grant number 03EI1004A-E. The authors highly appreciate the support of Andreas Meurer for the visualization of data. We would like to thank Aileen Böhme for developing and supporting the scenario driver, Manuel Wetzel for his support with REMix and PIPS-IPM++, and Kai von Krbek, Sonja Simon and Mengzhu Xiao for their contributions to the indicator processing. The authors gratefully acknowledge the Gauss Centre for Supercomputing e.V. (www.gauss-centre.eu) for funding this project by providing computing time through the John von Neumann Institute for Computing (NIC) on the GCS Supercomputer JUWELS at Jülich Supercomputing Centre (JSC).

## Author contributions

U.J.F, K.-K.C., S.S., J.B. and T.B. drafted the manuscript. U.J.F, S.S., J.B. and T.B. did data curation and implemented required model modifications. U.J.F, K.-K.C. and S.S. conducted the investigation process. U.J.F and K.-K.C. conceptualized the study, acquired funding and finalized the manuscript. K.-K.C. and T.B. acquired the computing resources. U.J.F prepared the visualization of data and conducted the formal analyses. K.-K.C. designed the methodology and coordinated the research activities, T.B. implemented the software for HPC.

## Funding

## Competing interests

The authors declare no competing interests.
