## [Transparent Peer Review file · Nature Communications]

The benefits of exploring a large scenario space for future energy systems

Corresponding Author: Dr Ulrich Frey

Version 0:

Reviewer comments:

Reviewer #1

(Remarks to the Author)

In the manuscript, the authors discuss a large-scale study of comparing a very high number of scenario runs (around 11.000) using different models (energy system optimization model and agent-based electricity market model) and modelling/parameter choices. The underlying work is impressive and provides a very rich set of scenario data for analysis, allowing to study the impact of various modelling choices on the scenario results. Given the relevance of scenario results for policy discussions and as context for other research, such a study is highly relevant and suitable for a journal like Nature Communications. Presenting and discussing such work is very challenging though. The large number of scenario runs, modelling choices, parameters and indicators demand very clear definition of the specific research question and focus in the presentation and discussion of the analysis. Also the level of detail in describing the models is challenging, because some results may depend on specific modelling details which need to be explained to the reader, while the complexity of the models does not allow to give a complete review of all applied modelling choices.

We thus acknowledge both the importance and quality of the underlying work and the challenges in presenting the analysis in the scope of a journal paper. Nevertheless, in the current form the article fails to present a clear discussion of the given research questions. We recommend a major revision to clarify the scope of the work and to restructure the presentation, so the findings of the study are understandable to the reader and learnings for the community can be drawn.

In the following we discuss some points in a more detailed fashion.

Abstract:

The scope and context of the research is unclear. "Energy scenario analysis" is a very broad term, what type of studies are meant? Also expressions like "minimal number of runs necessary" are very unclear, "minimal" for what purpose?

Introduction:

Although the sub-headline "Energy scenario analysis should think big" is catchy, its relevance and meaning for the given presentation is unclear. The discussion in this (brief) section is superficial and fails to provide the context and scope for the given article.

The research questions given in lines 68 to 71 are specific and the very brief overview in lines 72 to 74 gives a good idea about what is actually done in this study. The remaining introduction should focus on putting this into context and discussing how these specific research questions can contribute to scenario analysis.

Some more specific remarks for this part:

Line 20: What exactly is meant by "verifiability" here?

Lines 20 - 23: A discussion how and why these points are related to a limited scenario space is missing

Line 27: Why and how do "running thousands of scenarios can overcome the above limitations"?

Lines 37 and 51-53: In line 37 the authors set the focus on energy system optimization models, later in 51-53 they give some examples for using only a few scenarios. Are the given examples all based on ESOMs? And aren't these very different types of scenarios (target scenarios vs. projections) based on very different models, so how relevant is the limited scenario space for all of them?

Line 55: What does "plausible" mean here exactly?

Main:

We understand that the underlying models cannot be explained in detail in the main part. But nevertheless, in the main part *some* model description should be given, in particular if the following presentation of scenario results contains details like emissions, generation capacities, etc. We recommend to briefly describe the type of models and where modelling decisions and parameters vary, including a more detailed presentation in the appendix.

Also, a discussion of the different *types* of variations in scenario setting is missing. Input parameter variations might be due to uncertainties or different literature sources, a decision about brownfield or greenfield might be due to the specific research question. Also the choice of parameter and modelling decision variations is not discussed - Why were this specific choice made?

Furthermore, the scope of the underlying model should be explained. If it only contains the power sector, how is the demand for it modelled? What about increasing electricity demand from the heating, transport and industrial sector? How relevant are modelling decisions about sectoral integration or potential model coupling in comparison with the sensitivities applied in this study?

Figure 1:

In scenario analysis emissions are often introduced as a boundary constraint ("emission cap") - it is interesting to observe that emissions vary here, due to the penalty factor briefly mentioned somewhere in the appendix. The modelling choice to *not* use an emission cap but rather a penalty factor should be discussed here. Also the temporal scope of the scenarios is unclear, given that the Climate Action Law gives a temporal evolution of emissions in the power sector (with a non-trivial translation in an overall budget). And do we consider CO₂ or CO₂eq here? And what is the relevance of the UBA 2017 emission budget?

Line 103: We do not understand the reference to the legend of Figure 1. Also the "grey scenario" as a benchmark should be explained in the main text.

Figure 3:

The figure is quite difficult to understand and should feature a) better description in the figure itself (explain which Colour is normal, and which truncated Distribution, explain the small tables in the figure and what the Symbols mean) b) better explanation in the text

The discussion in (III.) gives a good example of the shortcomings of the current presentation. There are modelling choices about the available generation technologies for different settings, but the argument for these choices and the impact on the results are not discussed. Also the potential reasoning for greenfield vs. brownfield are not discussed.

The presentation in (IV.) acknowledges that there has been work on the sensitivity of the results of ESOMs on the choice of weather years, but no effort is done to relate the findings in the literature to the results presented by the authors.

Line 199: It remains unclear, what the criteria for "robust, trustworthy results" are.

Discussion:

Also here, the arguments are unclear. How exactly does the work prove the "need to consider different method choices and multiple indicators beyond costs"? For which purpose do we need to consider which choices how? What is relation to the overall choice of modelling class, to sector integration, to pathway vs. greenfield vs. brownfield, etc.? Later it is stated that "the last percentage points of decarbonization become increasingly cost intensive", but how is this statement based on the analysis? Or is it just a finding from the general literature rephrased here?

Scenario generator and parameter sampling:

We as reviewers did not check all sources to judge comprehensibility of the models used in this study. Some points which remain unclear considering the discussion in the appendix in the given manuscript:

In line 294 "expert interviews" are mentioned, but it remains unclear how exactly this process entered into scenario parameter determination. Maybe this is reviewed in another source?

In line 325 it is casually mentioned that 20% of the started runs crashed due to not fully understood circumstances. Here a slightly more elaborated discussion would be helpful - is this a computational detail or relevant for the overall scope of the research?

In line 335ff. the spatial aggregation is described. Usually, different levels of spatial aggregation in ESOMs refer to different clustering levels. Here, a part of the system is always kept highly resolved - why? How does this affect the results?

In "Models" the usage of REMIX as a framework for ESOMs is reviewed and the alternative/additional(?) usage of AMIRIS introduced. It remains unclear what exactly the role of AMIRIS is. From the description it appears that AMIRIS does not include investment decisions, so how is this market operation coupled or independently run with/from an ESOM?

Reviewer #2

(Remarks to the Author)

Reviewer #3

(Remarks to the Author)

The article "The benefits of exploring a large scenario space" proposes to investigate a large number of energy system scenarios, based both on system optimization and agent-based models. The authors use HPC systems to achieve the large number of simulations (hundreds to thousands) and study how much the scenarios differ in terms of some defined key indicators.

The article is mostly well-written, following a clear line of thought with sound analysis. Still, there are several comments to be addressed before publication:

1. Given that the main contribution is setting-up of an HPC system to generate a large number of scenarios, I have two concerns:

- a) The runtime/computational complexity is not disclosed. How much compute was needed?
- b) No code is openly shared. This severely limits the reproducibility and added value for any researcher trying to generate more scenarios themselves.

2. "The technical details of the implementation are published in 37."

I found reference 37 very expansive, containing many results, including the HPC workflow as well as additional indicators. I would ask the authors to clarify the contributions and differences of the current paper compared to prior work.

3. I have some concerns about the different scenarios:

- a) Single (historic) weather years might be insufficient given the expected change in weather due to climate change.
- b) n-1 vulnerability is considered. However, in the future, we expect the grid to be expanded substantially. Are grid extensions considered?

4. Pareto fronts are mentioned but not shown. I think this or any alternate usage of the large scenario count would substantially increase the added value of the paper.

5. The language, e.g. in the abstract, is sub-optimal: it should be more clearly distinguished between motivation, research gap and the contribution by the present paper/the authors.

6. There is some confusion about the indicators: "4 out of 6 indicators" and "seven key indicators": This is inconsistent and confusing

Reviewer #4

(Remarks to the Author)

The paper studies two sophisticated energy system models given a distribution of scenarios and various modeling choices. It evaluates the outcome distribution using several indicators.

The paper highlights the need to use a broad range of scenarios and evaluate the results using appropriate measures.

In principle, I believe the paper could be interesting and might be suitable for publication in Nature Communications. It addresses an important point: The robust analysis of model properties for several modeling choices under broad uncertainties. However, I believe the presentation is lacking. Due to the lack of precision and details, many aspects of the results are hard/impossible to understand or judge accurately. Further, the analysis stops short of some interesting questions, and the results are not investigated deeply enough.

If my points can be addressed, I would recommend publication.

Novelty

It remains somewhat unclear what the exact contribution of the paper is. They refer several times to established consensus that several hundred scenarios are required, but fail to clearly state how their scenario process differs, especially at the modeling level.

My impression is that a key contribution is the automated scenario generation tool that encodes the results of a literature review and expert interviews. Unfortunately, this is not easily usable/accessible to other researchers. The authors should consider publishing at least the scenario tool (and the underlying data) as an independent software package. I believe that from a modeling perspective, this is a key contribution. As is, the methods of the paper are not easily usable by other researchers.

If (as they note in the introduction) the authors view the HPC pipeline, and the analysis of larger ensembles of model runs using indicators are the strongest contribution, then I think open-source publishing the entire pipeline as a usable package would be appropriate. In that case, I would especially want to see a more detailed analysis of the resulting outcome ensembles.

Details of the scenario generation process and model inputs

Related to the above: The authors describe how their scenario generation process was designed, but the scenario generation process is lacking in details. The authors also don't explain the models used, even at a conceptual level. Thus,

for example, it is unclear what the model inputs are for which distributions are assumed. (Reference 38 does not clarify this, I tracked down some information in reference 37, but this isn't too useful either). This makes it difficult to accurately judge several important points:

* I am puzzled by the statement that the mean load varies dramatically between different modeling choices (page 5 line 137). Usually, I would expect loads to be inputs, and uncertainties to not alter the mean.

* The precise distributions used are not given. It is stated that from literature research median, min and max are obtained. Presumably a uniform distribution between min and max is chosen. But how is the truncated normal built? How is the variance chosen?

* Indeed, my strong impression is that the variance of the uniform distribution is simply much larger than that of the truncated normal used. In this case, it is not the modeling choice of normal vs uniform, but the variance that drives the strong divergence of the results.

* Related to this is the interpretation of min/max in the interviews/literature: Is this the largest value people estimate as a mean or the largest value people think could plausibly occur?

Properties of the scenario space

The authors give various marginals of the outcome of their experiments, but it would also be highly interesting to explore the scenario space a bit more thoroughly. Figure 3 is convincing to me and would deserve more elaboration. Studying the minimum number of runs needed does not strike me as crucially important in comparison.

I would suggest adding pair plots (in the sense of <https://seaborn.pydata.org/generated/seaborn.pairplot.html>) that show the relationship among outcome variables (and maybe some key input variables), at least in the SI. This would demonstrate how the scenario variations correlate, and which input uncertainties drive this. The only place where the question of which input uncertainties drive the results is discussed is in the context of the loads, which, as noted above, is puzzling.

Presentational points:

I also consider the presentation of the paper to be lacking, especially given the broader audience of Nature Communications.

1) The authors don't introduce the models used accurately enough. I do not mean a lack of technical information, but even at a conceptual level. I would expect that the introduction would explain something along the lines of "We use two models, the first is a linear optimization model (REMIX) that takes x, y, z as input and provides us with a, b, c . This type of model is used to answer questions such as..., we then take the output of this model, and further inputs x', y', z' as the input of a second model, which is of type w and is used to answer questions such as...". It is unclear what the constraints for the model are.

2) The introduction also unnecessarily conflates a number of actually quite different ways in which uncertainties appear in different contexts, making it hard to follow. The authors are doing Monte Carlo to understand the impact of uncertainties on model outputs. This is straightforward, and it is not necessary or helpful to cite but not really explain other contexts in which stochastic programming or MGA are used. MGA especially does something quite fundamentally different at a conceptual level. (There are also language problems in the introduction specifically which do not aid in understanding "... has to be addressed caused by ..." in line 56 page 2. The entire paragraph from line 54 to 64 does not really add to the paper, in my view).

3) The authors don't differentiate very different types of methodological choices. When it comes to spatial resolution, or greenfield vs brownfield, we know that one is more realistic than the other. For normal vs truncated, we do not know this. For network vulnerability, it is also unclear to me how realistic their approach is, and what it implies exactly.

Conclusions

The conclusions drawn are too cursory and not easy to follow given the results. It is unclear why in line 230 Pareto frontiers are mentioned, as my understanding is that this work presented no Pareto frontier. It is also unclear why in line 234 the authors claim that these results confirm that close to the cost optimum we have many results that perform heterogeneously. Unless I fundamentally misunderstood the paper, there was no modeling of alternatives or near optimal states here. Instead, it was shown that under different parameters, the cost optimum can look very differently. This is of course somewhat spiritually related, but the two should not be conflated.

For several of their modeling choices, the authors don't really draw a conclusion (e.g. Network vulnerabilities) or only confirm expected results (more than one weather year is required) without showing in detail (e.g. in the SI) what goes wrong otherwise. Generally, the resulting distribution of scenarios and results is not investigated very deeply. The authors don't investigate what input uncertainties actually drive the observed variation in indicators. They don't investigate how outcome variations correlate, and they don't ask the question of how these relationships might differ given different modeling choices. I believe these types of questions can be answered with a relatively minor investment of work, and they might make the paper suitable for publication in Nature Communications.

Reviewer #5

(Remarks to the Author)

Reading this manuscript, I found myself agreeing with many of the points raised: the need for running thousands of

scenarios for robust energy system planning, the challenges in presenting results of a large scenario space, and the need and skills for using high-performance computing for reliable insights.

However, while the range of presented scenarios is impressive and commendable, in my opinion, substantial improvements would have to be made for the manuscript to be suitable for publication in Nature Communications. I have concerns about the translation of the results into actionable insights, how general the insights might be for alternative model setups, as well as the level of detail of the methodology description and the motivation for certain assumptions/analyses. My impression is that the model opens up many result dimensions across a wide range of methodological and techno-economic assumptions, without discussing their implications in detail and offering only limited discussion of them.

Unclear target audience and policy relevance:

To me, it is not quite clear what the target audience of the article is. It seems to be directed at other energy system modellers, as some parts are too technical for a broader audience or policy relevance. On the other hand, if the target audience are other modellers, some sections, e.g. "Why energy systems research struggles ...", are too basic. Also, while the example of the German power system in 2030 is illustrative for the chosen modelling approach, it is not particularly relevant because it - as far as I could tell - omits anticipated developments in terms of coupling to other sectors (electric vehicles, heat pumps, etc.).

Lack of detail in model description:

The model description should include more detail to understand the model results. A short overview in the introduction section would also be helpful. While many features of the tools are listed that are *not* used in the study, a lot of information and motivation for many assumptions is missing. For instance: In the spatial resolution scenarios, why is only NRW highly resolved and the rest of Germany not? Why cluster according to electrical distance? What is the range of CO2 costs considered? Why are power plants considered only until 2018? Shouldn't this data be up to date for such a study rather than six years old, especially in terms of installed wind and solar capacities? How are the seven indicators developed?

For the scenarios, the authors consider a mix of methodological and parametric choices, relating to techno-economic uncertainties, spatial resolution, legacy infrastructure and weather variability. While the range of considered uncertainties is broader than in many other studies, its selection is not comprehensibly motivated. Couldn't this also lead to some bias or at least not very general results? The mix of methodological and parametric choices also makes it difficult to disentangle the effect of model simplifications from real-world uncertainties about costs.

I am also not convinced that the claims made in the section "Minimum number of runs needed for unbiased results" are general enough. Besides the desired confidence levels, doesn't the required number of runs also depend on the number of uncertainties considered and the general model structure/type? I could imagine that the numbers would look quite a bit different for another model with sector-coupling looking at net-zero emission systems. How were the 3000 scenarios chosen as an upper limit and are 100 samples enough to remove bias?

Unclear motivation for model coupling:

It is not very clear why two models (REMIX and Amiris) are needed to make the argument about the importance of exploring a large scenario space. The authors make the point about two modelling approaches, "decentralized actors" and "central planner", but overall my impression is that it adds unnecessary complication to the experimental setup. It would be helpful if the authors could clarify the motivation for this setup to support their argument.

Detailed comments:

Title: Short is good, but since it was submitted to a journal with very broad topic coverage, it needs to state somehow that it is about energy systems.

Abstract: In my opinion, the abstract focuses too much on the methods and results, but does not motivate the study very well or highlight the main takeaway messages. The summary of results is also quite unspecific and hard to understand initially without having read the paper.

P1-L25: Given that there are many studies with dozens to hundreds of energy scenarios evaluated on HPC infrastructure (e.g. [https://www.cell.com/joule/fulltext/S2542-4351\(22\)00236-7](https://www.cell.com/joule/fulltext/S2542-4351(22)00236-7), [https://www.cell.com/joule/fulltext/S2542-4351\(23\)00266-0](https://www.cell.com/joule/fulltext/S2542-4351(23)00266-0)), I would suggest to rephrase this statement. Using HPC is the state-of-the-art and established best practice -- it is more a question of how many papers apply it.

P2-L44: Perhaps change the wording from "impractical" to something like "challenging". Using "impractical" sounds like large-scale simulations are not possible at all.

P2-L51: All of the named examples on small scenario space are not referring to ESOMs, are they? Here, one could refer to a actually few policy relevant examples where the number of scenarios is small but billions of dollars are put on the line (e.g. TYNDP of ENTSO-E and ENTSO-G).

P2-L62: Why only near-term actions? Stochastic approaches could also guide long-term decisions.

Figure 1: UBA may not be known to those unfamiliar with German energy policy.

Figure 1: Is the network vulnerability scenario not covered by this figure?

Figure 2: Doesn't the averaging dilute some of the differences within the groups of method choice? Focusing on distributions for each generation type relative to the reference group (e.g. one panel each) would increase the information content. It would also avoid mixing RES capacities with conventional capacities, which are difficult to compare.

Distributions and sampling: Is it the normal versus uniform distribution per se that causes this difference or is it rather how the ranges and cut-off points of the normal distribution are chosen? How is the truncation done for the normal distribution? Does this have a big influence on the results? More information on what input parameters are varied should be given. Otherwise it is difficult to understand the results and put them in perspective. The parameter space should not be relegated to an external source. It is also unclear who the interviews were conducted with and what sources were used.

The impact of spatial resolution is found to be almost negligible. How can this be aligned with other studies with opposite indications (e.g. <https://doi.org/10.1016/j.apenergy.2021.116726>, <https://doi.org/10.1016/j.esr.2020.100563>, <https://doi.org/10.1016/j.esr.2019.100362>)?

Could you explain why scenario reduction techniques would reduce the information content in your case by so much? Also, there are for the random drawing from the distributions there are low-discrepancy sampling techniques that can cover the uncertainty space more quickly. Couldn't these have been chosen to reach faster convergence?

The discussion is currently quite shallow. The first paragraph still presents further results and the latter two, while insightful, only present a list of limitations. The results should be put into a broader context in relation to actionable insights for the target audience.

Code availability: It is unclear why certain parts of the code are not made available.

P9-L282: It is not quite clear what the purpose of this paragraph is.

Version 1:

Reviewer comments:

Reviewer #1

(Remarks to the Author)

The revised manuscript by Frey et al. addresses the comments made in the previous review. The article now is much more focussed and understandable for the reader, outlining the scenario creation methods and the underlying models.

We think the research is interesting for readers of the journal and deserves publication. Before publication, we would recommend to clearer communicate the scope and limits of the analysis. To our understanding, the main point of the article is that energy system models necessarily involve model building choices (methodological uncertainties) and uncertainties related to data (parameteric uncertainty) (line 90). The authors present a workflow which allows to scan a large scenario space involving variations in both these dimensions, allowing to provide robust results over such a large sample.

As stated in the article, the analysis of the modelling choices and parameter variations never can be complete - there are always more parameter variations and modelling choices to explore, like alternative weather years, integration of more sectors, or additional modeling dimensions like endogenous learning. It would be worthwhile to discuss the potential and limitations of the presented workflow more generally. For instance, after line 360 the authors discuss limitations of the study, first addressing sector coupling. Could the workflow presented in the papers be adapted to a sector-coupled model? Would it make sense to compare a sector-coupled with a electricity sector only model inside this workflow?

To our understanding the authors consider different modelling and parametric choices for representing "the same system" using structurally similar models (for the energy system optimization part) - is this a correct understanding of the limits of the work flow? We recommend to discuss this clearer in the discussion section, maybe also making the scope of this work clearer already in the introduction.

Once the focus is made clearer and potentials/limitations are discussed, we recommend publication. Additionally, the article should be re-checked, in the PDF version available to the reviewers, several references are broken.

(Remarks on code availability)

Reviewer #2

(Remarks to the Author)

(Remarks on code availability)

Reviewer #3

(Remarks to the Author)

I thank the authors for their extensive revision and answers to the comments raised by me and the other reviewers. With the scope clarified and new results added, I believe this is a suitable contribution to Nature Communications.

(Remarks on code availability)

Reviewer #4

(Remarks to the Author)

The paper is much improved, and many concerns, especially regarding the software stack and description of model and modeling choices have been addressed well.

I maintain that many of the differences seen between truncated normal and uniform distribution are likely due to the larger variance of the latter, but at least now the process by which the distributions were chosen is clearly explained.

In my view the simulated ensemble looks really interesting, but its properties are still not sufficiently explored in the paper. However, I am happy to defer to other reviewers on this, as this is a bit outside my core area of expertise.

(Remarks on code availability)

Reviewer #5

(Remarks to the Author)

I have reviewed the revised materials, and while I find that some aspects could be cleared up and improved, unfortunately, many of the issues I criticised in the initial submission persist.

My impression is that for the millions of CPU hours expended, the generated new insights are relatively low -- or at least discussed with too little depth. Based on the research questions raised (usefulness of HPC and influence methodological choices), first of all, I feel that the target audience is relatively narrow (i.e., other modellers) and not the broad audience interested in energy systems analysis. Second, the contributions mentioned (raising awareness about uncertainties and the impact of method choices) are not exactly news to other modellers. To most, it will already be clear that large scenario spaces are required. Furthermore, while this manuscript may provide the first combined analysis of different method choices, ample literature examples individually evaluate these aspects (e.g., weather years, spatial resolution, and techno-economic assumptions). It has not been demonstrated what the added value of the combination of all of these factors is, and it seems to me that this may have come at the cost of a detailed discussion of the effects and implications. Also, I would reiterate that the example of the German power system in 2030 (in 5 years) might not be the most suitable choice for illustrating the effects, as anticipated future developments in technology deployment will aggravate many of the effects analysed (e.g. uncertainty in cost developments, dependence on weather variability, effect of grid bottlenecks).

It has not been a major deciding factor in my assessment, but I also found the responses to the reviews often too sparse, not describing specifically what adjustments were made. Some comments were cursorily dismissed as being "out of scope" or with reference to a word limit that is not strictly imposed. The improvements in the methodology description are small, and the need for AMIRIS in the workflow is still not convincingly motivated.

I apologise for being so critical in the second round of reviews; while I appreciate the improvements in the current revision, my overall judgement is that this revised manuscript is unsuitable for publication in Nature Communications and might be better suited for submission to a more technical journal.

(Remarks on code availability)

Version 2:

Reviewer comments:

Reviewer #5

(Remarks to the Author)

I appreciate the authors' efforts in refining their manuscript in previous revisions. I can see some improvements, and I think they match what can be reasonably expected from the authors. While I am happy to be overruled by the other reviewers, my final recommendation would be to accept the manuscript for publication in a more specialised journal. The reason I do not recommend this manuscript for publication in Nature Communications is that, in my view, the fundamental criticisms regarding the narrow target audience, novelty of insights, coherence of the paper's storyline, clarity of the study design, and relevance of the case study outlined in previous reviews remain. That said, I do think the manuscript has merit and would be of interest to a more specialised audience.

(Remarks on code availability)

Dear reviewers,

thank you very much for your very detailed and constructive comments and suggestions.

Below you will find all our answers to them. We are confident that the manuscript is now much improved thanks to your very valuable remarks!

Reviewer #1 (Remarks to the Author):

In the manuscript, the authors discuss a large-scale study of comparing a very high number of scenario runs (around 11.000) using different models (energy system optimization model and agent-based electricity market model) and modelling/parameter choices. The underlying work is impressive and provides a very rich set of scenario data for analysis, allowing to study the impact of various modelling choices on the scenario results. Given the relevance of scenario results for policy discussions and as context for other research, such a study is highly relevant and suitable for a journal like Nature Communications. Presenting and discussing such work is very challenging though. The large number of scenario runs, modelling choices, parameters and indicators demand very clear definition of the specific research question and focus in the presentation and discussion of the analysis. Also the level of detail in describing the models is challenging, because some results may depend on specific modelling details which need to be explained to the reader, while the complexity of the models does not allow to give a complete review of all applied modelling choices.

We thus acknowledge both the importance and quality of the underlying work and the challenges in presenting the analysis in the scope of a journal paper. Nevertheless, in the current form the article fails to present a clear discussion of the given research questions. We recommend a major revision to clarify the scope of the work and to restructure the presentation, so the findings of the study are understandable to the reader and learnings for the community can be drawn.

In the following we discuss some points in a more detailed fashion.

Abstract:

1. The scope and context of the research is unclear. "Energy scenario analysis" is a very broad term, what type of studies are meant? Also expressions like "minimal number of runs necessary" are very unclear, "minimal" for what purpose?

We reworked the abstract completely, taking especial care to explain better what type of studies are meant. The sentence with the "minimal number of runs" has been deleted and rewritten completely.

Introduction:

Although the sub-headline "Energy scenario analysis should think big" is catchy, its relevance and meaning for the given presentation is unclear. The discussion in this (brief) section is superficial and fails to provide the context and scope for the given article.

We deleted the sub-headline and replaced it with a better headline. We also provide much more context and scope in the introduction and also refer the readers to the next section, where we also added more context.

The research questions given in lines 68 to 71 are specific and the very brief overview in lines 72 to 74 gives a good idea about what is actually done in this study. The remaining introduction should focus on putting this into context and discussing how these specific research questions can contribute to scenario analysis.

We now include two paragraphs about our contribution, i.e. increasing the robustness of ESOM approaches by analyzing the influence of five frequent method choices on results. In addition, the introduction has been completely rewritten.

Some more specific remarks for this part:

Line 20: What exactly is meant by "verifiability" here?

Since the introduction has been completely rewritten, this line is no longer there, but replaced by a clearer formulation.

Lines 20 - 23: A discussion how and why these points are related to a limited scenario space is missing

Thank you for your comment. We agree that the first section of the introduction was not very clear and therefore we reformulated this section. We now differentiate between different aspects of uncertainty that we try to tackle within this paper.

Line 27: Why and how do "running thousands of scenarios can overcome the above limitations"? We added an illustrating example, i.e. using 24 weather years instead of just one.

Lines 37 and 51-53: In line 37 the authors set the focus on energy system optimization models, later in 51-53 they give some examples for using only a few scenarios. Are the given examples all based on ESOMs? And aren't these very different types of scenarios (target scenarios vs. projections) based on very different models, so how relevant is the limited scenario space for all of them?

We have changed the references to show that even in studies that depend on large investments, no comparable parameter variations are analysed.

Line 55: What does "plausible" mean here exactly?

This is now explained.

Main:

We understand that the underlying models cannot be explained in detail in the main part. But nevertheless, in the main part *some* model description should be given, in particular if the following presentation of scenario results contains details like emissions, generation capacities, etc. We recommend to briefly describe the type of models and where modelling decisions and parameters vary, including a more detailed presentation in the appendix.

We included a short introduction to the energy system optimization model REMix and to the agent-based model AMIRIS at the end of the introduction and at the beginning of the main section. A more detailed description of the models and the parameter variations can be found in the “Methods” section. We also added a new figure (Figure 1) to illustrate the workflow. Furthermore, we included Table 2 in the appendix where the values for the parameter variations are listed.

Also, a discussion of the different *types* of variations in scenario setting is missing. Input parameter variations might be due to uncertainties or different literature sources, a decision about brownfield or greenfield might be due to the specific research question.

We appreciate this comment and considered it in the revised version of the manuscript by clearly distinguishing parametric and methodological uncertainties. We argue that both are related to decisions that are made by modelers. However, while parametric uncertainties are at least somehow addressed in state-of-the-art modeling approaches, the systematic evaluation of what we call method choices is rather lacking. Moreover, we added a paragraph on the selection of method choices to the Discussion section, where we admit that the investigation of further method choices shall be subject to future research.

Also the choice of parameter and modelling decision variations is not discussed - Why were this specific choice made?

This was based on a broad literature review. This review identified the main parameters known to be important for the ESOM. We tried to be as comprehensive as possible. We also explain this better in the text and the SI.

Furthermore, the scope of the underlying model should be explained. If it only contains the power sector, how is the demand for it modelled? What about increasing electricity demand from the heating, transport and industrial sector? How relevant are modelling decisions about sectoral integration or potential model coupling in comparison with the sensitivities applied in this study? Our parameter variation considers the increase in annual demand with minimum, median and maximum values (see Table 1 in the SI). The demand consists of power demand from the power sector, but also of power demand from the transport and heating sector. The demand profile is based on historic years and can be sampled independent from the annual demand. This information is now added to the section “Scenario generation and parameter sampling”.

Sector integration is key in a highly decarbonized future energy system since it can provide additional flexibility options. However, we consider a partially decarbonized power system in 2030. The consideration of sector integration was out of scope for our project. This has been added to the discussion. Especially the spatial resolution method choice could have a higher impact on the optimization results in a highly decarbonized energy system with sector integration.

Figure 1:

In scenario analysis emissions are often introduced as a boundary constraint (“emission cap”) - it is interesting to observe that emissions vary here, due to the penalty factor briefly mentioned somewhere in the appendix. The modelling choice to *not* use an emission cap but rather a penalty factor should be discussed here.

Budgets for CO₂ emissions are often discussed in politics, but so far, they are rather used as benchmark than a strict limitation. By sampling the CO₂ emission price as one of many parameter uncertainties we can analyze the effect of different CO₂ prices in combination with other parameter variations on the cost and carbon emissions for each modeling run. This has also been added to section "Scenario space for the German power system".

Also the temporal scope of the scenarios is unclear, given that the Climate Action Law gives a temporal evolution of emissions in the power sector (with a non-trivial translation in an overall budget).

Thank you, this was unclear but is clearly very important. We added at several places that 2030 is the target year.

And do we consider CO₂ or CO₂eq here? And what is the relevance of the UBA 2017 emission budget?

We clarified the question on CO₂ or CO₂eq in the paragraph above the corresponding figure. Moreover, by also considering further review comments, we added a disclaimer about the contributions of our study, where we state that we cannot providing scenarios directly useable for policy-advice. One of the reasons for this is the application of a target year planning approach for the power sector only. Therefore, the directly reportable CO₂ emissions are very limited in their scope. However, we want to provide some context information, which is why we identified UBA 2017 as a reference having the same scope and thus, provides values for CO₂ emissions useful for a fair contextualization.

Line 103: We do not understand the reference to the legend of Figure 1. Also the "grey scenario" as a benchmark should be explained in the main text.

This is now explained much better and in detail by having added various sections, also for the "grey scenario benchmark".

Figure 3:

The figure is quite difficult to understand and should feature a) better description in the figure itself (explain which Colour is normal, and which truncated Distribution, explain the small tables in the figure and what the Symbols mean) b) better explanation in the text

Yes, this is true. We have now improved the figure in various ways: first, the legends (small tables) have now the scenario names as short code in them, making it easy to match the characteristics to the respective scenario. We have also moved the figure out of the "distribution section" up to the text of method choices, so as to avoid confusion with that particular methodological choice. We also refer the scenario names to the SI table 2 and explain the figure in much more detail in the text.

The discussion in (III.) gives a good example of the shortcomings of the current presentation. There are modelling choices about the available generation technologies for different settings, but the argument for these choices and the impact on the results are not discussed. Also the potential reasoning for greenfield vs. brownfield are not discussed.

Thank you for the comment. We agree that a more detailed analysis and discussion on the method

choices is beneficial. We therefore extended this section and discuss also further conceivable method choices.

The presentation in (IV.) acknowledges that there has been work on the sensitivity of the results of ESOMs on the choice of weather years, but no effort is done to relate the findings in the literature to the results presented by the authors.

Thank you for your comment. We included a comparison of our results to findings in the literature where also an impact on the expanded capacities can be observed when analyzing several weather years. The results suggest that the impact can be rather limited if fewer weather years are considered.

Line 199: It remains unclear, what the criteria for "robust, trustworthy results" are.

As mentioned in the text. This depends on the researcher's questions and use cases. There cannot be one definition.

Discussion:

Also here, the arguments are unclear. How exactly does the work proof the "need to consider different method choices and multiple indicators beyond costs"? For which purpose do we need to consider which choices how?

We find that method choices do have an important influence on results, but it is not always to use one instead of another.

Weather years: results are clearly more robust with more weather years.

Distribution: normal distributions are more realistic, while uniform distributions makes extreme cases more frequent. This means that calculated future energy systems become more robust.

Brownfield vs. greenfield: brownfield systems are more realistic again, but greenfield systems allow to research configurations that you would never see with a brownfield approach, allowing us to demonstrate alternatives.

So, depending on your goal, different method options are available. However, researchers should be aware of their influence – which is exactly what we try to show in this paper.

What is relation to the overall choice of modelling class, to sector integration, to pathway vs. greenfield vs. brownfield, etc.?

Unfortunately, these questions are out of scope of this paper, given that we are now at 5000 words.

However, we now briefly mention these questions, e.g. sector integration.

Later it is stated that "the last percentage points of decarbonization become increasingly cost intensive", but how is this statement based on the analysis? Or is it just a finding from the general literature rephrased here?

We deleted this sentence, but also added a figure in the SI, Figure 1, showing all pairwise comparisons.

Scenario generator and parameter sampling:

We as reviewers did not check all sources to judge comprehensibility of the models used in this study. Some points which remain unclear considering the discussion in the appendix in the given manuscript:

In line 294 "expert interviews" are mentioned, but it remains unclear how exactly this process entered into scenario parameter determination. Maybe this is reviewed in another source?

Thank you for this comment. We extended the description of our approach in "Scenario generation and parameter sampling".

In line 325 it is casually mentioned that 20% of the started runs crashed due to not fully understood circumstances. Here a slightly more elaborated discussion would be helpful - is this a computational detail or relevant for the overall scope of the research?

We extended the sentence. It could also have been a numerical instability in PIPS++, but we can rule out memory issues. However, this is not relevant for the results, hence the "casual mention".

In line 335ff. the spatial aggregation is described. Usually, different levels of spatial aggregation in ESOMs refer to different clustering levels. Here, a part of the system is always kept highly resolved - why? How does this affect the results?

We included a much more detailed description within the "Spatial aggregation" section. We chose to model a technology-rich and highly interconnected region within Germany with high spatial resolution. This allows us to analyze different types of conventional power plants and their phase-out more precisely. Furthermore, the highly resolved part enables us to consider a set of different security of supply indicators and the unavailability of transformer substations.

In "Models" the usage of REMIX as a framework for ESOMs is reviewed and the alternative/additional(?) usage of AMIRIS introduced. It remains unclear what exactly the role of AMIRIS is. From the description it appears that AMIRIS does not include investment decisions, so how is this market operation coupled or independently run with/from an ESOM?

AMIRIS is important for the market indicators, one of the three groups of indicators. It also serves as plausibility check for the optimization model, e.g. extremely high electricity prices. Within the workflow, it is coupled to the results of REMIX. We describe its function now more clearly in the paper.

Reviewer #2 (Remarks to the Author):

Thank you for this in-depth review, which clearly improved the paper a lot!

Reviewer #3 (Remarks to the Author):

The article “The benefits of exploring a large scenario space” proposes to investigate a large number of energy system scenarios, based both on system optimization and agent-based models. The authors use HPC systems to achieve the large number of simulations (hundreds to thousands) and study how much the scenarios differ in terms of some defined key indicators.

The article is mostly well-written, following a clear line of thought with sound analysis. Still, there are several comments to be addressed before publication:

1. Given that the main contribution is setting-up of an HPC system to generate a large number of scenarios, I have two concerns:

a) The runtime/computational complexity is not disclosed. How much compute was needed?

Thank you for pointing this out. We added a paragraph in the text and a table with the details of the computational time in the SI (Table 3).

b) No code is openly shared. This severely limits the reproducibility and added value for any researcher trying to generate more scenarios themselves.

All elements of the workflow (AMIRIS, REMix, ioproc, PIPS, the scenario generator and JUBE) are in fact openly available. This is now referenced in the paper. Here is a list of all repositories:

<https://gitlab.com/dlr-ve/esy/remix/framework/>

<https://gitlab.com/dlr-ve/esy/amiris/amiris>

<https://github.com/PIPS-IPMpp/PIPS-IPMpp>

<https://github.com/FZJ-JSC/JUBE>

<https://gitlab.com/dlr-ve/esy/ioproc>

<https://gitlab.com/dlr-ve/esy/remix/projects/unseen>

2. “The technical details of the implementation are published in 37.”

I found reference 37 very expansive, containing many results, including the HPC workflow as well as additional indicators. I would ask the authors to clarify the contributions and differences of the current paper compared to prior work.

The mentioned reference is a technical report and rather grey literature than a scientific paper. We mainly describe our methods and workflow there without providing a dedicated scientific contribution. In contrast to the mentioned reference, this paper analyzes a multitude of scenario runs and results in order to provide an insight on the impact of different methodological choices, taking parameter uncertainties into account within a Monte Carlo simulation.

3. I have some concerns about the different scenarios:

a) Single (historic) weather years might be insufficient given the expected change in weather due to climate change.

Yes, you are right. This is one of the reasons to consider the method choices with multiple vs. one weather year. Although the impact of several weather years on the structure of the energy system is

not always considered in energy system optimization, weather is an important driver for the energy system as shown in section “Methodological choices” (IV.).

b) n-1 vulnerability is considered. However, in the future, we expect the grid to be expanded substantially. Are grid extensions considered?

Yes, by capacity expansion of existing transmission lines. The effect of the outage causes unavailability of certain connections for grid expansion, too.

4. Pareto fronts are mentioned but not shown. I think this or any alternate usage of the large scenario count would substantially increase the added value of the paper.

We totally agree and have thus included a section and a plot about our Pareto front results.

5. The language, e.g. in the abstract, is sub-optimal: it should be more clearly distinguished between motivation, research gap and the contribution by the present paper/the authors.

We have rewritten the abstract to better distinguish its sections. In the paper, we follow the requirements of the journal.

6. There is some confusion about the indicators: “4 out of 6 indicators “ and “seven key indicators”: This is inconsistent and confusing

Thank you. This is now corrected.

Reviewer #4 (Remarks to the Author):

The paper studies two sophisticated energy system models given a distribution of scenarios and various modeling choices. It evaluates the outcome distribution using several indicators.

The paper highlights the need to use a broad range of scenarios and evaluate the results using appropriate measures.

In principle, I believe the paper could be interesting and might be suitable for publication in Nature Communications. It addresses an important point: The robust analysis of model properties for several modeling choices under broad uncertainties. However, I believe the presentation is lacking. Due to the lack of precision and details, many aspects of the results are hard/impossible to understand or judge accurately. Further, the analysis stops short of some interesting questions, and the results are not investigated deeply enough.

If my points can be addressed, I would recommend publication.

Novelty

It remains somewhat unclear what the exact contribution of the paper is. They refer several times to established consensus that several hundred scenarios are required, but fail to clearly state how their scenario process differs, especially at the modeling level.

We completely revised our contribution statement in the introduction of our paper, and rewrote large parts of the introduction. We furthermore sharpened our contribution by highlighting our studies' benefits for model-based decision-support and for modelers/scientists that aim at providing it.

My impression is that a key contribution is the automated scenario generation tool that encodes the results of a literature review and expert interviews. Unfortunately, this is not easily usable/accessible to other researchers. The authors should consider publishing at least the scenario tool (and the underlying data) as an independent software package. I believe that from a modeling perspective, this is a key contribution. As is, the methods of the paper are not easily usable by other researchers.

All codes developed in the project are either already open source or have been published on <https://gitlab.com/dlr-ve/esy/remix/projects/unseen> to address the reviewer's concern. The basic parameters of the used REMix model instance, PowGER, has been openly published on <https://doi.org/10.23728/B2SHARE.A4E0822F80BB48B7A62106253B4189C0> and all scalar values of sampled REMix inputs and finally calculated indicators are published with [10.23728/b2share.7dfe93339c3e4e34bf4c47f880186466](https://doi.org/10.23728/b2share.7dfe93339c3e4e34bf4c47f880186466).

If (as they note in the introduction) the authors view the HPC pipeline, and the analysis of larger ensembles of model runs using indicators are the strongest contribution, then I think open-source publishing the entire pipeline as a usable package would be appropriate. In that case, I would especially want to see a more detailed analysis of the resulting outcome ensembles.

All elements of the workflow are published open source (see list above). However, the workflow itself is specific to the software stack of the High-performance-cluster in Juelich. It is also too data-intensive (see SI, table 3) with 40 Terabyte, to being contained in a useable package. On the other hand, we refer readers interested in reproducing our analyses to the extensive technical report (footnote 43, Anderson et al.) which provides a detailed description of the steps to follow.

Details of the scenario generation process and model inputs

Related to the above: The authors describe how their scenario generation process was designed, but the scenario generation process is lacking in details. The authors also don't explain the models used, even at a conceptual level. Thus, for example, it is unclear what the model inputs are for which distributions are assumed. (Reference 38 does not clarify this, I tracked down some information in reference 37, but this isn't too useful either). This makes it difficult to accurately judge several important points:

We included a short introduction to the energy system optimization model REMix and to the agent-based model AMIRIS at the end of the introduction and at the beginning of the main section. A more detailed description of the models can be found in the "Methods" section. Furthermore, we included Table 2 in the SI where the parameter variations are listed.

* I am puzzled by the statement that the mean load varies dramatically between different modeling choices (page 5 line 137). Usually, I would expect loads to be inputs, and uncertainties to not alter the mean.

It's a sampling input as stated in the text: "The nodal demands are subject to parameter sampling". However, we amended the text to make it clearer.

* The precise distributions used are not given. It is stated that from literature research median, min and max are obtained. Presumably a uniform distribution between min and max is chosen. But how is the truncated normal built? How is the variance chosen?

We extended our description of the truncated normal distribution in the section "Scenario generation and parameter sampling" and explained how the standard deviation is calculated.

* Indeed, my strong impression is that the variance of the uniform distribution is simply much larger than that of the truncated normal used. In this case, it is not the modeling choice of normal vs uniform, but the variance that drives the strong divergence of the results.

For both the uniform and truncated normal distribution we use the same minimum and maximum values. However, the probability distributions differ. This is described now in more detail in the section "Scenario generation and parameter sampling".

* Related to this is the interpretation of min/max in the interviews/literature: Is this the largest value people estimate as a mean or the largest value people think could plausibly occur?

Thanks. We added an explaining paragraph.

Properties of the scenario space

The authors give various marginals of the outcome of their experiments, but it would also be highly interesting to explore the scenario space a bit more thoroughly. Figure 3 is convincing to me and would deserve more elaboration. Studying the minimum number of runs needed does not strike me as crucially important in comparison.

We agree with the reviewer's recommendation to better elaborate the findings from the scenario

space exploration. For this, we extended the discussion of Figure 4 (formerly Figure 3). However, we are convinced that from a practical point of view the number of runs needed is still important since it determines the computational resource used. To more clearly communicate this point, we additionally discuss the need for such analyses in the Discussion, where we outline future research based on multi-objective optimization approaches.

I would suggest adding pair plots (in the sense of <https://seaborn.pydata.org/generated/seaborn.pairplot.html>) that show the relationship among outcome variables (and maybe some key input variables), at least in the SI. This would demonstrate how the scenario variations correlate, and which input uncertainties drive this. The only place where the question of which input uncertainties drive the results is discussed is in the context of the loads, which, as noted above, is puzzling.

Thank you for this valuable suggestion. We added pair plots for all indicators in the SI (Figure 1), so that correlations become much clearer.

Presentational points:

I also consider the presentation of the paper to be lacking, especially given the broader audience of Nature Communications.

1) The authors don't introduce the models used accurately enough. I do not mean a lack of technical information, but even at a conceptual level. I would expect that the introduction would explain something along the lines of "We use two models, the first is a linear optimization model (REMIX) that takes x, y, z as input and provides us with a, b, c . This type of model is used to answer questions such as..., we then take the output of this model, and further inputs x', y', z' as the input of a second model, which is of type w and is used to answer questions such as...". It is unclear what the constraints for the model are.

We included a short introduction to the energy system optimization model REMIX and to the agent-based model AMIRIS at the end of the introduction and at the beginning of the main section. With respect to the broader audience of Nature Communications we limited it to high-level statements. We provide an even more detailed description of the models in the "Methods" section. In addition to the already openly available modeling frameworks REMIX and AMIRIS, we also published the complete source code used to establish our modeling workflow.

2) The introduction also unnecessarily conflates a number of actually quite different ways in which uncertainties appear in different contexts, making it hard to follow. The authors are doing Monte Carlo to understand the impact of uncertainties on model outputs. This is straightforward, and it is not necessary or helpful to cite but not really explain other contexts in which stochastic programming or MGA are used. MGA especially does something quite fundamentally different at a conceptual level. (There are also language problems in the introduction specifically which do not aid in understanding "... has to be addressed caused by ..." in line 56 page 2. The entire paragraph from line 54 to 64 does not really add to the paper, in my view).

We agree to the reviewer's comment on MGA. The corresponding section has been revised and the MGA part is deleted from the introduction.

3) The authors don't differentiate very different types of methodological choices. When it comes to spatial resolution, or greenfield vs brownfield, we know that one is more realistic than the other.

For normal vs truncated, we do not know this. For network vulnerability, it is also unclear to me how realistic their approach is, and what it implies exactly.

We chose our method choices according to the literature. We are aware that these choices are very different concerning their "realism". However, the focus is not on how realistic they are, but to quantify the influence of one method choice in the context of other method choices. The goal is to help modelers to know which method choices are important and whether and how much they will influence their results. We completely rewrote the introduction and added various sentences throughout the paper to make this clearer.

Conclusions

The conclusions drawn are too cursory and not easy to follow given the results. It is unclear why in line 230 Pareto frontiers are mentioned, as my understanding is that this work presented no Pareto frontier.

You are right. We have expanded the section on Pareto frontiers and added a new figure (Figure 6) that sums up our results for this aspect of our work.

It is also unclear why in line 234 the authors claim that these results confirm that close to the cost optimum we have many results that perform heterogeneously. Unless I fundamentally misunderstood the paper, there was no modeling of alternatives or near optimal states here. Instead, it was shown that under different parameters, the cost optimum can look very differently. This is of course somewhat spiritually related, but the two should not be conflated.

You are right. We deleted the sentence.

For several of their modeling choices, the authors don't really draw a conclusion (e.g. Network vulnerabilities) or only confirm expected results (more than one weather year is required) without showing in detail (e.g. in the SI) what goes wrong otherwise.

You are right. We have expanded the corresponding text to address this objection.

Generally, the resulting distribution of scenarios and results is not investigated very deeply. The authors don't investigate what input uncertainties actually drive the observed variation in indicators. They don't investigate how outcome variations correlate, and they don't ask the question of how these relationships might differ given different modeling choices. I believe these types of questions can be answered **with a relatively minor investment of work**, and they might make the paper suitable for publication in Nature Communications.

We agree that a more detailed analysis of the input uncertainty and their impact on the indicators is important. While this is out of scope for the presented work since the focus is mainly on the method choices, we analyzed the impact on indicators in a previous publication (<https://doi.org/10.3389/frevo.2024.1398358>). As stated before, we added Supplementary Figure 1, where the interrelation between the indicators is shown for the reference case. Furthermore, we included Supplementary Table 11 and Supplementary Table 12 where the mean and standard deviation of power plant capacities are listed for the considered scenario ensembles indicating the impact of the parameter uncertainties and method choices.

Reviewer #5 (Remarks to the Author):

Reading this manuscript, I found myself agreeing with many of the points raised: the need for running thousands of scenarios for robust energy system planning, the challenges in presenting results of a large scenario space, and the need and skills for using high-performance computing for reliable insights.

However, while the range of presented scenarios is impressive and commendable, in my opinion, substantial improvements would have to be made for the manuscript to be suitable for publication in Nature Communications.

I have concerns about the translation of the results into actionable insights, how general the insights might be for alternative model setups, as well as the level of detail of the methodology description and the motivation for certain assumptions/analyses. My impression is that the model opens up many result dimensions across a wide range of methodological and techno-economic assumptions, without discussing their implications in detail and offering only limited discussion of them.

Unclear target audience and policy relevance:

To me, it is not quite clear what the target audience of the article is. It seems to be directed at other energy system modellers, as some parts are too technical for a broader audience or policy relevance. On the other hand, if the target audience are other modellers, some sections, e.g. "Why energy systems research struggles ...", are too basic. Also, while the example of the German power system in 2030 is illustrative for the chosen modelling approach, it is not particularly relevant because it - as far as I could tell - omits anticipated developments in terms of coupling to other sectors (electric vehicles, heat pumps, etc.).

We appreciate this comment as it follows internal discussions about the target audience. As our target journal is Nature Communications, we have a claim to provide insights for a broad audience. However, we acknowledge that we have sometimes failed to explain why it is important to be aware of the technical details presented. Therefore, we have completely rewritten the introduction to better motivate our idea of raising awareness about technical details such as the method choices investigated.

In addition, we clarify the objective of the study by adding disclaimers to avoid misunderstandings regarding the provision of new energy scenarios for German policymakers. We have rewritten the conclusions accordingly. Finally, we have added the limited accuracy of the used energy system models regarding sector coupling as a topic in the discussion section and emphasize our expectations regarding the effects we expect from modeling the corresponding additional interdependencies.

Lack of detail in model description:

The model description should include more detail to understand the model results. A short overview in the introduction section would also be helpful. While many features of the tools are listed that are *not* used in the study, a lot of information and motivation for many assumptions is missing. For instance: In the spatial resolution scenarios, why is only NRW highly resolved and the rest of Germany not? Why cluster according to electrical distance? What is the range of CO2 costs considered? Why are power plants considered only until 2018? Shouldn't this data be up to date for such a study rather than six years old, especially in terms of installed wind and solar capacities? How are the seven indicators developed?

We included a more detailed description within the “Spatial aggregation” section, where we explain why we chose to cluster according to electrical distance and to keep one part of the model with high spatial resolution. Furthermore, we included Table 1 in the SI where the parameter variations, such as the CO2 price, are listed.

The project started already in 2019. This was also the time when we started to collect data and did our first modeling runs. Since we wanted to analyze the impact of the different method choices, we decided to stick with the initially collected data in order to be able to compare earlier results with results conducted at a later point in time within the project. However, the conclusions of this paper on the impact of the method choices should remain the same also with more recent data. We used the data on existing capacities as a lower limit for our optimization. However, the upper limit for the expansion of renewable energies was chosen according to the analysis of land use data and therefore much higher. This is also described in section “Capacity expansion approach”.

Our goal was to analyze many different aspects when we assess the energy system scenarios. Therefore, for this paper we chose seven out of 34 collected indicators that represent the aspects of affordability, security and sustainability.

For the scenarios, the authors consider a mix of methodological and parametric choices, relating to techno-economic uncertainties, spatial resolution, legacy infrastructure and weather variability. While the range of considered uncertainties is broader than in many other studies, its selection is not comprehensibly motivated. Couldn't this also lead to some bias or at least not very general results? The mix of methodological and parametric choices also makes it difficult to disentangle the effect of model simplifications from real-world uncertainties about costs.

The considered parameter uncertainties are based on a broad literature review. For every techno-economic parameter where different values were found in the literature, the values are sampled within our Monte Carlo analysis. This insures that the selection of sampled parameters is not biased. The interrelation matrix is based on a workshop where experts from the field of energy system analysis were interviewed. For the method choices we picked five that are frequently used within energy system optimization but where the impact of different choices has not systematically been analyzed.

I am also not convinced that the claims made in the section "Minimum number of runs needed for unbiased results" are general enough. Besides the desired confidence levels, doesn't the required number of runs also depend on the number of uncertainties considered and the general model structure/type? I could imagine that the numbers would look quite a bit different for another model with sector-coupling looking at net-zero emission systems. How were the 3000 scenarios chosen as an upper limit and are 100 samples enough to remove bias?

We repeated the run shown 10 times with this scenario group. In addition, to address your criticism, we ran the analysis for other scenario groups as well. Both result sets are very similar, resulting in the exact same ranking of indicators, indicating that the results are indeed robust.

The number 3000 was chosen because this was the number of the largest scenario group calculated. This being our largest sample, it is used in order to find out about reducing bias as much as possible.

Furthermore, also other studies indicate that “the number of runs required is independent of the number of uncertain parameters, and mainly depends on the level of confidence” [<https://www.sciencedirect.com/science/article/pii/S2211467X18300543>].

Unclear motivation for model coupling:

It is not very clear why two models (REMix and Amiris) are needed to make the argument about the importance of exploring a large scenario space. The authors make the point about two modelling approaches, "decentralized actors" and "central planner", but overall my impression is that it adds unnecessary complication to the experimental setup. It would be helpful if the authors could clarify the motivation for this setup to support their argument.

We answer that question above now in more detail. The main reason is that each model is able to calculate its indicators best, since it is specialized for them. The technical paper also explains this in much detail (Reference 43, Anderson et al.).

Detailed comments:

Title: Short is good, but since it was submitted to a journal with very broad topic coverage, it needs to state somehow that it is about energy systems.

Thank you. We changed the title.

Abstract: In my opinion, the abstract focuses too much on the methods and results, but does not motivate the study very well or highlight the main takeaway messages. The summary of results is also quite unspecific and hard to understand initially without having read the paper.

We provide a completely revised abstract where we rebalanced the share of statements concerning motivation, methods, results and impact.

P1-L25: Given that there are many studies with dozens to hundreds of energy scenarios evaluated on HPC infrastructure (e.g. [https://www.cell.com/joule/fulltext/S2542-4351\(22\)00236-7](https://www.cell.com/joule/fulltext/S2542-4351(22)00236-7), [https://www.cell.com/joule/fulltext/S2542-4351\(23\)00266-0](https://www.cell.com/joule/fulltext/S2542-4351(23)00266-0)), I would suggest to rephrase this statement. Using HPC is the state-of-the-art and established best practice -- it is more a question of how many papers apply it.

Thank you, you are right. It is now rephrased.

P2-L44: Perhaps change the wording from "impractical" to something like "challenging". Using "impractical" sounds like large-scale simulations are not possible at all.

Replaced.

P2-L51: All of the named examples on small scenario space are not referring to ESOMs, are they? Here, one could refer to a actually few policy relevant examples where the number of scenarios is small but billions of dollars are put on the line (e.g. TYNDP of ENTSO-E and ENTSO-G).

Thank you for this suggestion. We have changed the references accordingly.

P2-L62: Why only near-term actions? Stochastic approaches could also guide long-term decisions.

You are right. We added long-term decisions.

Figure 1: UBA may not be known to those unfamiliar with German energy policy.

It is now explained.

Figure 1: Is the network vulnerability scenario not covered by this figure?

You are right. We decided to leave them out in this figure, since they are – compared to the other scenario groups – very small ($n = 100$). For the goal of this figure, they do not contribute enough to merit their inclusion. However, we analyse the effects of network vulnerabilities later on, see e.g. Figure 4.

Figure 2: Doesn't the averaging dilute some of the differences within the groups of method choice? Focusing on distributions for each generation type relative to the reference group (e.g. one panel each) would increase the information content. It would also avoid mixing RES capacities with conventional capacities, which are difficult to compare.

Thank you for your comment. We are in favor of keeping the bars averaged since they provide a better overview of the impact of the method choices.

However, we agree that the distribution per technology and scenario would be an added value. Therefore, we now include Supplementary Table 11 and Supplementary Table 12 where the mean and standard deviation of power plant capacities are listed for the considered scenario ensembles indicating the impact of the parameter uncertainties and method choices.

Distributions and sampling: Is it the normal versus uniform distribution per se that causes this difference or is it rather how the ranges and cut-off points of the normal distribution are chosen? How is the truncation done for the normal distribution? Does this have a big influence on the results? More information on what input parameters are varied should be given. Otherwise it is difficult to understand the results and put them in perspective. The parameter space should not be relegated to an external source. It is also unclear who the interviews were conducted with and what sources were used.

Thank you for your comment. We added a more detailed description on how the truncated normal distribution is calculated based on the minimum, maximum and median values of our parameters. Additionally, we added table 1 in the SI, where the values for our parameter variation are listed and extended section "Scenario generation and parameter sampling" to give a better overview on the considered uncertainties. The interviews were conducted with experts in the field of energy system analysis.

The impact of spatial resolution is found to be almost negligible. How can this be aligned with other studies with opposite indications (e.g. <https://doi.org/10.1016/j.apenergy.2021.116726>, <https://doi.org/10.1016/j.esr.2020.100563>, <https://doi.org/10.1016/j.esr.2019.100362>)?

The differences in results compared to previous studies can be explained by two key aspects. First, our study employs clustering based on electrical distance, whereas existing studies primarily use clustering methods based on renewable energy generation potential and k-means. Second, our findings suggest that while there is an impact of different cluster sizes, the impact is relatively minor compared to the influence of other methods examined in our study. Frysztacki et al. show that while the total power plant capacity over all technologies does not change to a high extent, wind offshore gets replaced by wind onshore and that higher costs are mainly driven by network bottlenecks. This is similar to the findings in our paper. Additionally, while other studies focus mainly on system costs as the primary comparison indicator, our analysis considers a significantly broader set of indicators.

Could you explain why scenario reduction techniques would reduce the information content in your case by so much? Also, there are for the random drawing from the distributions there are low-discrepancy sampling techniques that can cover the uncertainty space more quickly. Couldn't these have been chosen to reach faster convergence?

We appreciate this comment. The idea of scenario reduction is to yield a representative sub-set of the initial scenario set while maintaining the important information. In general, similar scenarios are grouped with each other. The similarity assessment is based on the input parameters. In our specific case, we sample two different types of input parameters: i) less than 100 scalar techno-economic values and ii) time series of different weather years consisting of at least 8760 interrelated values per region and technology. Due to the unbalance of these two parameter types, scenario reduction would result in scenarios grouped by weather years, while undesirably losing the variation of the techno-economic parameters. We did not further investigate more sophisticated sampling techniques.

The discussion is currently quite shallow. The first paragraph still presents further results and the latter two, while insightful, only present a list of limitations. The results should be put into a broader context in relation to actionable insights for the target audience.

Thank you. We agree with this comment and accordingly extended the discussion of results for the broad readership of the journal.

Code availability: It is unclear why certain parts of the code are not made available.

Please see our reply (list of repositories) above. All elements of the workflow are openly available.

However, it needs to be noted that the developed HPC workflow and the associated software is a mixture of applying established software packages, newly developed software and platform specific single-use scripts. Therefore, we first did not see a benefit in publishing the software packages since full reproducibility is only possible when having the domain knowledge of the different research disciplines involved in its development. In other words, we assume that extensive support would be necessary to enable full use. However, we agree with the reviewer that for the sake of transparency all software and data shall be accessible. Therefore, we provide both the full set of input and output data as open data and all newly developed software as open source.

P9-L282: It is not quite clear what the purpose of this paragraph is.

It is now rephrased and much more concrete.

Rebuttal

Dear reviewers, dear Dr. Xue,

thank you very much for your again very constructive comments.

Below you will find all our answers in green. We do hope that the manuscript is now ready for publication! We also implemented all the formatting requirements, as requested.

Reviewer 1 +2:

The revised manuscript by Frey et al. addresses the comments made in the previous review. The article now is much more focused and understandable for the reader, outlining the scenario creation methods and the underlying models.

Thank you.

Before publication, we would recommend to clearer communicate the scope and limits of the analysis. As stated in the article, the analysis of the modelling choices and parameter variations never can be complete - there are always more parameter variations and modelling choices to explore, like alternative weather years, integration of more sectors, or additional modeling dimensions like endogenous learning. It would be worthwhile to discuss the potential and limitations of the presented workflow more generally. For instance, after line 360 the authors discuss limitations of the study, first addressing sector coupling. Could the workflow presented in the papers be adapted to a sector-coupled model? Would it make sense to compare a sector-coupled with a electricity sector only model inside this workflow?

We added more information on the scope and also the limitations of the workflow to the Discussion.

To our understanding the authors consider different modelling and parametric choices for representing "the same system" using structurally similar models (for the energy system optimization part) - is this a correct understanding of the limits of the work flow? We

recommend to discuss this clearer in the discussion section, maybe also making the scope of this work clearer already in the introduction.

Yes, this is a correct understanding. We added a further explanation in the introduction and the discussion about the scope.

Additionally, the article should be re-checked, in the PDF version available to the reviewers, several references are broken.

Thank you. We checked all references. However, on our end, every reference links to the correct article. Yet, we will re-check once we will be in the copy-editing stage. We added missing DOIs for 8 of them.

Reviewer 3:

I thank the authors for their extensive revision and answers to the comments raised by me and the other reviewers. With the scope clarified and new results added, I believe this is a suitable contribution to Nature Communications.

Response: Thank you for your time and effort!

Reviewer 4:

The paper is much improved, and many concerns, especially regarding the software stack and description of model and modeling choices have been addressed well.

I maintain that many of the differences seen between truncated normal and uniform distribution are likely due to the larger variance of the latter, but at least now the process by which the distributions were chosen is clearly explained.

In my view the simulated ensemble looks really interesting, but its properties are still not sufficiently explored in the paper. However, I am happy to defer to other reviewers on this, as this is a bit outside my core area of expertise.

Thank you for this valuable feedback. Following your suggestion, we tested the variances across all scenarios and the uniform distributions do have a larger variance than the normal distributions. This results in a wider spread of system costs as shown in Fig. 4 (I). Since the

truncated normal distribution is mostly skewed to the right (see Supplementary Table 1), extremely high costs and high demand have a rather low probability (in comparison to extremely low costs and demand). Therefore, the system costs with a uniform and a truncated normal distribution applied do not share the same mean value. The uniform distribution results in higher system costs than the truncated normal distribution. We have added this information in section “Methodological choices” (I).

Reviewer 5:

I feel that the target audience is relatively narrow (i.e., other modelers) and not the broad audience interested in energy systems analysis.

Thank you for pointing this out again. We are aware that this is a balancing act between the specific modeling topics we address and the conclusions for a broader audience. We already rewrote large parts of the paper for the first revision, particularly the Introduction and the Conclusion to include topics for a broader audience.

In addition, we are convinced of demonstrating to a broad audience the importance of the limitations of modelling decisions that are often made intuitively. For this reason, we are determined to continue with the chosen journal. We are confident that the latest revisions, made possible by your comments, successfully convey this idea.

We also revised statements in the Introduction and the Discussion again. We did this to keep the perspective of non-modelling experts. In addition, we highlighted information we consider relevant specifically for this target group. Furthermore, we now explicitly state the target in group the Discussion we would like to reach.

Second, the contributions mentioned (raising awareness about uncertainties and the impact of method choices) are not exactly news to other modelers. To most, it will already be clear that large scenario spaces are required.

This is undoubtedly true. However, this has not yet resulted in approaches that actually include the uncertainties introduced into modeling by method choices. Therefore, this paper demonstrates quantitatively for a very large number of scenarios that these impacts can be quite large and cannot be addressed with just citing references about uncertainties.

In contrast, we argue that they have to be incorporated into the modeling workflow in energy systems analysis in general. As mentioned above, our aim is to provide our message not only

to modelers but to convey the dangers of method choice to the scientific community as a whole (which is one point interesting for a broader audience, by the way).

This is why the paragraph you are referring to addresses non-modelers. Being modelers ourselves, the trade-off between revealing novel insights to both of the target groups is hard to find. For example, we are aware that the value of a large scenario space is clear to the majority of modelers, yet it is not state of the art. This is why we argue in our manuscript that there are several reasons why scanning large scenario spaces is not yet the standard. In addition, modelers often neglect uncertainties in their analysis. They often consider them in very limited sensitivity analyses, if at all. At the same time, they are aware of their relevance. Therefore, with our comprehensive and quantifiable assessment, we provide help on which uncertainties to focus, since such exercises are out of scope for most studies.

As a consequence, we adapted the introduction and discussion to more clearly state the intended target groups of our contributions.

Furthermore, while this manuscript may provide the first combined analysis of different method choices, ample literature examples individually evaluate these aspects (e.g., weather years, spatial resolution, and techno-economic assumptions). It has not been demonstrated what the added value of the combination of all of these factors is, and it seems to me that this may have come at the cost of a detailed discussion of the effects and implications.

You are perfectly right – there is literature about individual evaluations of method choices.

However, we assert that our study allows for the first time a *comparison* of these effects since it analyzes the effects of method choice on *the same data set*. For example, while previous research has also highlighted the impact of the spatial resolution on the energy system, our study shows that the impact on the structure of the energy system is rather low in comparison to the impact of other method choices (see Fig. 3). Moreover, this data set is a substantial set of scenarios (11,000+) making our claims robust – and with all software components published as open-source – also reproducible for the community. While spatial effects are often analyzed, our paper gains relevance through the comparison of method choices.

Furthermore, we do not only analyze the impact of each model choice on the structure (Fig. 3) and the total cost of the energy system but we also consider the impact on a broad set of

indicators (Fig. 4) representing the affordability, security and sustainability of the optimized energy system (see Table 1).

We modified the Introduction to highlight the added value.

Also, I would reiterate that the example of the German power system in 2030 (in 5 years) might not be the most suitable choice for illustrating the effects, as anticipated future developments in technology deployment will aggravate many of the effects analyzed (e.g. uncertainty in cost developments, dependence on weather variability, effect of grid bottlenecks).

We agree with the reviewer's assessment.

We apologize for a misunderstanding in the last revision, where we accidentally removed the promised disclaimer in the Introduction.

If our study's objective is to provide insights for deriving actions to be taken to transform the current energy system within five years, the chosen target year would be an appropriate choice. The revised disclaimer makes this clear. We initially hesitated to include a specific target year in the manuscript's first version, but we took the comments of reviewer #1 seriously and included the explicit statement of a target year of 2030 in our first revision. However, we are aware that justifying this 2030-label is complicated by the fact that we sampled almost all parameters that characterize a scenario year. Existing power plants, projected fuel prices, investments, and normative targets for climate change mitigation are examples of the relevant parameters. However, they have been varied within large ranges. This may have led to parameter sets that are more plausible for a 2035 scenario.

Finally, we explain the decision not to target overly ambitious scenarios. We made this decision during the project runtime (in 2021) because we wanted to keep the uncertainty of a changing regulatory framework manageable for the simulation of the power market. This reflects our doubt that in systems with large shares of renewable energies the (energy only-) market design will persist.

We removed the sentence "It provides plausible scenarios for Germany's future power system infrastructure operated under real market conditions" from the abstract to prevent confusion with scenario studies to be directly used for policy advice. In addition, we added the disclaimer to avoid wrong expectations from the results of the use case studied.

It has not been a major deciding factor in my assessment, but I also found the responses to the reviews often too sparse, not describing specifically what adjustments were made. Some comments were cursorily dismissed as being "out of scope" or with reference to a word limit that is not strictly imposed.

We are very sorry to hear that you felt that our answers were too sparse. We are trying to improve this in this round of reviews.

Concerning "out of scope", we felt that our paper is already touching on too many aspects for one paper. Take for example the Pareto front analysis that had previously been done by us, but had been deleted in the first version of the paper for exactly this reason. Since a reviewer requested it, we included it once more. While a valuable contribution, it adds another tangent off the main focus of the paper.

Moreover, before publication and in each round of reviews we have been reminded by the editor of the strict requirements concerning word limit, indicating that non-compliance would result in "we will not be able to proceed with your revised manuscript otherwise". Hence, we have to work under the assumption that the word limit is strictly enforced.

The improvements in the methodology description are small, and the need for AMIRIS in the workflow is still not convincingly motivated.

Thank you for this comment. We try to make the need for AMIRIS in the workflow clearer again. REMix optimizes the energy system from a holistic central planner's perspective with perfect foresight to find the optimal expansion and dispatch of the energy system. In comparison, AMIRIS simulates each participant of the electricity market as one prototypical actor. Each actor has its own strategy, trying to maximize its profit without taking the optimal operation of the energy system into account. Therefore, by applying AMIRIS electricity market driven indicators can be considered in our workflow.

We improved the description of AMIRIS in the section "Models – AMIRIS". Furthermore, in a previous paper from our group (<https://doi.org/10.1016/j.egy.2022.12.145>) we describe the differences between REMix and AMIRIS in even more detail – this paper is now added as a reference.